# Kindlin-2 cooperates with talin to activate integrins and induces cell spreading by directly binding paxillin

Marina Theodosiou[1], Moritz Widmaier[1], Ralph T Böttcher[1], Emanuel Rognoni[1], Maik Veelders[1], Mitasha Bharadwaj[2], Armin Lambacher[1], Katharina Austen[1], Daniel J Müller[2], Roy Zent[3,4], Reinhard Fässler[1]*

[1]Department of Molecular Medicine, Max Planck Institute of Biochemistry, Martinsried, Germany; [2]Department of Biosystems Science and Engineering, Eidgenössische Technische Hochschule Zürich, Basel, Switzerland; [3]Division of Nephrology, Department of Medicine, Vanderbilt University, Nashville, United States; [4]Department of Medicine, Veterans Affairs Medical Center, Nashville, United States

**Abstract** Integrins require an activation step prior to ligand binding and signaling. How talin and kindlin contribute to these events in non-hematopoietic cells is poorly understood. Here we report that fibroblasts lacking either talin or kindlin failed to activate $\beta$1 integrins, adhere to fibronectin (FN) or maintain their integrins in a high affinity conformation induced by $Mn^{2+}$. Despite compromised integrin activation and adhesion, $Mn^{2+}$ enabled talin- but not kindlin-deficient cells to initiate spreading on FN. This isotropic spreading was induced by the ability of kindlin to directly bind paxillin, which in turn bound focal adhesion kinase (FAK) resulting in FAK activation and the formation of lamellipodia. Our findings show that talin and kindlin cooperatively activate integrins leading to FN binding and adhesion, and that kindlin subsequently assembles an essential signaling node at newly formed adhesion sites in a talin-independent manner.

*For correspondence: faessler@biochem.mpg.de

## Introduction

Integrins are heterodimeric transmembrane receptors that mediate cell adhesion to the extracellular matrix (ECM) and to other cells (*Hynes, 2002*). The consequence of integrin-mediated adhesion is the assembly of a large molecular network that induces various signaling pathways, resulting in cell migration, proliferation, survival and differentiation (*Winograd-Katz et al., 2014*). The quality and strength of integrin signaling is controlled by the interaction between integrins and substrate-attached ligands, which is, in turn, regulated by the on- and off-rates of the integrin–ligand binding process. The on-rate of the integrin–ligand binding reaction (also called integrin activation or inside-out signaling) is characterized by switching the unbound form of integrins from an inactive (low affinity) to an active (high affinity) conformation. The affinity switch proceeds from a bent and clasped low affinity conformation to an extended and unclasped high affinity conformation with an open ligand-binding pocket (*Askari et al., 2010*; *Springer and Dustin, 2012*). This change in affinity is believed to be induced through the binding of talin and kindlin to the β integrin cytoplasmic domain (*Moser et al., 2009*; *Shattil et al., 2010*) and divalent cations to distinct sites close to the ligand-binding pocket (*Gailit and Ruoslahti, 1988*; *Mould et al., 1995*; *Xia and Springer, 2014*; *Mould et al., 2003*).

The stabilisation of integrin–ligand complexes is mediated by integrin clustering and catch bond formation between integrin and bound ligand. The stabilizing effect of clustered integrins is

**eLife digest** A meshwork of proteins called the extracellular matrix surrounds the cells that make up our tissues. Integrins are adhesion proteins that sit on the membrane surrounding each cell and bind to the matrix proteins. These adhesive interactions control many aspects of cell behavior such as their ability to divide, move and survive.

Before integrins can bind to the extracellular matrix they must be activated. Previous research has shown that in certain types of blood cells, proteins called talins and kindlins perform this activation. These proteins bind to the part of the integrin that extends into the cell, causing shape changes to the integrin that allow binding to the extracellular matrix. However, it is not clear whether talin and kindlin also activate integrins in other cell types.

Fibroblasts are cells that help to make extracellular matrix proteins, and are an important part of connective tissue. Theodosiou et al. engineered mouse fibroblast cells to lack either talin or kindlin, and found that both of these mutant cell types were unable to activate their integrins and as a result failed to bind to an extracellular matrix protein called fibronectin.

Even when cells were artificially induced to activate integrins by treating them with manganese ions, cells lacking talin or kindlin failed to fully activate integrins and hence did not adhere well to fibronectin. This suggests that talin and kindlin work together to activate integrins and to maintain them in this activated state.

When treated with manganese ions, cells that lacked talin were able to flatten and spread out, whereas cells that lacked kindlin were unable to undergo this shape change. Theodosiou et al. found that this cell shape is dependent on kindlin and its ability to bind to and recruit a protein called paxillin to "adhesion sites", where integrins connect the cell surface with the extracellular matrix. Kindlin and paxillin then work together to activate other signaling molecules to induce the cell spreading.

The next challenge is to understand how talin and kindlin are activated in non-blood cells and how they maintain integrins in an active state.

achieved by the ability of dissociated integrin–ligand complexes to rebind before they leave the adhesion site (*van Kooyk and Figdor, 2000*; *Roca-Cusachs et al., 2009*), while catch bonds are receptor–ligand bonds whose lifetime increases with mechanical force (*Kong et al., 2009*; *Chen et al., 2010*; *Kong et al., 2013*). Both mechanisms extend the duration and increase the strength of integrin-mediated adhesion and signaling (also called outside-in signaling) (*Koo et al., 2002*; *van Kooyk and Figdor, 2000*; *Maheshwari et al., 2000*; *Roca-Cusachs et al., 2009*; *Coussen et al., 2002*), and depend on the association of integrins with the actin cytoskeleton via talin (*Roca-Cusachs et al., 2009*; *Friedland et al., 2009*), and probably kindlin (*Ye et al., 2013*).

The talin family consists of two (talin-1 and -2) and the kindlin family of three isoforms (kindlin-1-3), which show tissue-specific expression patterns (*Calderwood et al., 2013*; *Moser et al., 2009*; *Shattil et al., 2010*). The majority of studies that defined integrin affinity regulation by talin and kindlin were performed on αIIbβ3 and β2-class integrins expressed by platelets and leukocytes, respectively. These cells circulate in the blood and hold their integrins in an inactive state until they encounter soluble or membrane-bound agonists (*Evans et al., 2009*, *Bennett, 2005*). The prevailing view is that agonist-induced signaling pathways activate talin-1 and the hematopoietic cell-specific kindlin-3, which cooperate to induce integrin activation (*Moser et al., 2008*; *Han et al., 2006*) and clustering (*Cluzel et al., 2005*; *Ye et al., 2013*).

Integrin affinity regulation in non-hematopoietic cells such as fibroblasts and epithelial cells is poorly understood. It is not known how integrin activation is induced on these cells (no integrin-activating agonists have been identified) and it is also controversial whether talin and kindlin are required to shift their integrins into the high affinity state. While there are reports showing that talin and kindlin are required for integrin activation in epithelial cells (*Montanez et al., 2008*; *Margadant et al., 2012*), it was also shown that in myoblasts and mammary epithelial cells activation of β1 integrins, adhesion and spreading on multiple ECM substrates can proceed in the absence of talin (*Conti et al., 2009*; *Wang et al., 2011*). Likewise, it was reported that focal adhesion

kinase (FAK)-deficient fibroblasts develop small, nascent adhesions (NAs) at the edge of membrane protrusions without visible talin and that integrins carrying a mutation in the talin-binding site can still nucleate and stabilize NAs (*Lawson et al., 2012*). Also fibroblasts lacking talin-1 and -2 were shown to adhere to fibronectin (FN) and initiate isotropic spreading (*Zhang et al., 2008*). Another intriguing study demonstrated that overexpression of kindlin-2 in Chinese hamster ovary (CHO) cells inhibits rather than promotes talin head-induced α5β1 integrin activation (*Harburger et al., 2009*). Given the fundamental importance of talin and kindlin for integrin activation in hematopoietic cells, the findings of these studies are unexpected and imply that either integrin affinity regulation is substantially different in fibroblasts and epithelial cells or the experimental approaches used to manipulate protein expression and localization were imperfect.

To directly evaluate the functions of talin and kindlins for FN-binding integrins on fibroblasts, we used a genetic approach and derived fibroblasts from mice lacking either the *Tln1* and *-2* or the *Fermt1* and *-2* genes. We show that integrin affinity regulation depends on both talin and kindlin, and that kindlin has the additional function of triggering cell spreading by binding directly to paxillin in a talin-independent manner.

## Results

### Kindlins and talins control cell morphology, adhesion and integrin expression

To obtain cells lacking the expression of talin-1 and kindlin-2, we intercrossed mice carrying *loxP* flanked (floxed; fl) *Tln1* or *Fermt2* alleles (*Figure 1A*), isolated kidney fibroblasts and immortalized them with the SV40 large T antigen (parental fibroblasts). The floxed alleles were deleted by adenoviral *Cre* recombinase transduction resulting in T1[Ko] and K2[Ko] fibroblasts. Loss of talin-1 or kindlin-2 expression in fibroblasts was compensated by talin-2 or the de novo expression of kindlin-1, respectively, allowing adhesion and spreading, although to a lesser extent compared with control cells (*Figure 1—figure supplement 1A,B*). To prevent this compensation, we generated mice with floxed *Tln1* and nullizygous *Tln2* alleles or with floxed *Fermt1* and *-2* alleles (Tln[Ctr]; Kind[Ctr]) from which we isolated, immortalized and cloned kidney fibroblasts with comparable integrin surface levels (*Figure 1A* and *Figure 1—figure supplement 2*). The floxed alleles were deleted by transducing *Cre* resulting in talin-1, *-2* (Tln[Ko]) and kindlin-1, *-2* (Kind[Ko]) deficient cells, respectively (*Figure 1A–C*). Since the Tln[Ctr] and Kind[Ctr] control cells showed similar morphologies and behaviour in our experiments, we display one control cell line in several result panels. *Cre*-mediated deletion of the floxed *Tln1* or floxed *Fermt1/2* genes was efficient (*Figure 1B*) and resulted in cell rounding, weak adhesion of a few cells, and reduced cell proliferation despite the immortalisation with the oncogenic large T antigen (*Figure 1C* and *Figure 1—figure supplement 3*). To minimize cell passage-induced abnormalities, we used cells only up to 12 passages after *Cre*-mediated gene deletions.

To define the adhesion defect, we performed plate and wash assays for 30 min on defined substrates and found that neither Tln[Ko] nor Kind[Ko] cells adhered to FN, laminin-111 (LN), type I collagen (COL) and vitronectin (VN) (*Figure 1D*). To test whether the inability of Tln[Ko] and Kind[Ko] cells to adhere to ECM proteins is due to an integrin activation defect, we bypassed inside-out activation by treating cells with $Mn^{2+}$, which binds to the integrin ectodomain and induces unbending and unclasping of integrin heterodimers (*Mould et al., 1995*). Treatment with $Mn^{2+}$ induced partial adhesion of Tln[Ko] and Kind[Ko] cells to FN, while partial adhesion to LN and VN was only induced in Tln[Ko] cells (*Figure 1D*). Time course experiments revealed that $Mn^{2+}$-induced adhesion of Tln[Ko] and Kind[Ko] cells to FN was already significantly lower 2.5 min after plating and remained significantly lower compared with control cells (*Figure 1E*), suggesting that talin and kindlin cooperate to initiate and maintain normal $Mn^{2+}$-induced adhesion to FN. In line with these findings, dose-response profiles showed that Tln[Ko] and Kind[Ko] cells have severe adhesion defects at low (1.25 μg ml$^{-1}$) as well as high (20 μg ml$^{-1}$) substrate concentrations (*Figure 1—figure supplement 4*).

These findings indicate that talin and kindlin promote integrin-mediated adhesion to FN and proliferation, and that the integrin-activating compound $Mn^{2+}$ can only partially substitute for the adhesion promoting roles that talin and kindlin accomplish together.

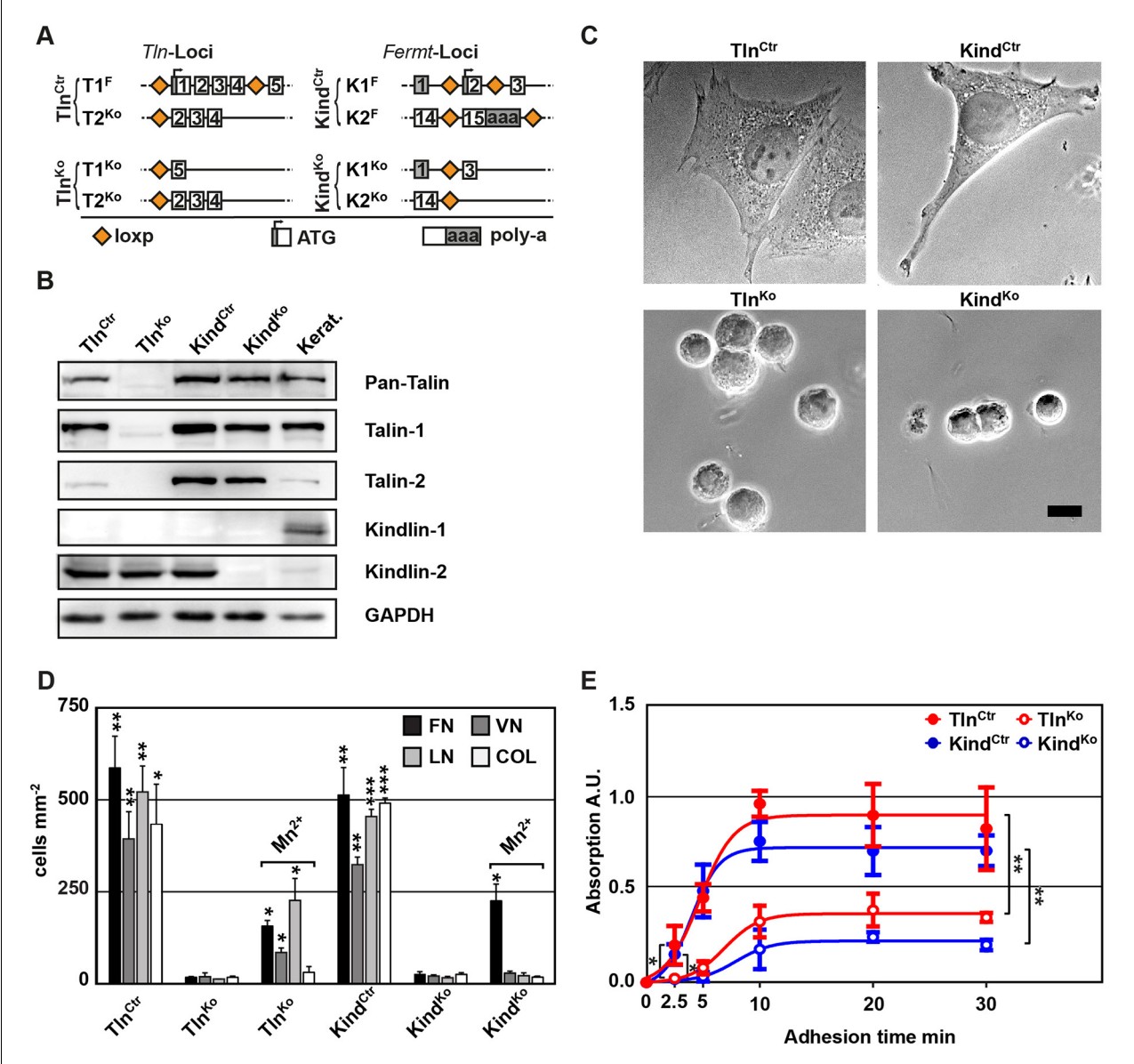

**Figure 1.** Kindlin and talin are required for integrin-mediated cell adhesion. (**A**) Scheme showing gene loci before and after ablation of the *Tln1, -2* and *Fermt1, -2* genes. Orange diamonds indicate *loxP* sites and rectangles exons; untranslated regions are marked grey. (**B**) Western blot of Tln[Ko] and Kind[Ko] cells. Keratinocyte lysates (Kerat.) served to control kindlin-1 expression. (**C**) Bright field images of Tln[Ctr], Kind[Ctr], Tln[Ko] and Kind[Ko] cells. (**D**) Quantification of cell adhesion on indicated substrates 30 min after seeding by counting DAPI stained cells; n=3 independent experiments, error bars indicate standard error of the mean; t-test significances are calculated between untreated Tln[Ko] or Kind[Ko] cells and the corresponding Tln[Ctr] and Kind[Ctr] or Mn[2+]-treated Tln[Ko] or Kind[Ko] cell lines on same substrates; only significant differences are shown. (**E**) Quantification of Mn[2+]-stimulated cell adhesion for indicated times on FN; cells were quantified by absorbance measurement of crystal violet staining; n=3 independent experiments; lines represent sigmoidal curve fit; error bars indicate standard deviation; significances for indicated pairs after 2.5 min were calculated by two-tailed t-test and significances for indicated pairs of the overall kinetics were calculated by two-way RM ANOVA. Bar, 10 μm. COL, collagen; DAPI, 4',6-diamidino-2-phenylindole; FN, fibronectin; GAPDH, glyceraldehyde-3-phosphate dehydrogenase; LN, laminin-111; RM ANOVA, repeated measures analysis of variance; VN, vitronectin.

The following figure supplements are available for figure 1:

**Figure supplement 1.** Talin-1- and kindlin-2-deficient fibroblasts.

**Figure supplement 2.** Integrin expression profiles of Tln[Ctr] and Kind[Ctr] cells.

*Figure 1 continued on next page*

**Figure supplement 3.** Cell proliferation of Tln$^{Ko}$ and Kind$^{Ko}$ cells.
DOI: 10.7554/eLife.10130.006
**Figure supplement 4.** Cell adhesion of Tln$^{Ko}$ and Kind$^{Ko}$ cells on different FN concentrations.
DOI: 10.7554/eLife.10130.007

## Integrin activation and binding to FN requires talin and kindlin-2

The inability of Mn$^{2+}$ to fully rescue the adhesion defect of Tln$^{Ko}$ and Kind$^{Ko}$ cells raised the question whether integrin surface levels change after deletion of the *Tln1/2* and *Fermt1/2* genes. We quantified integrin surface levels by flow cytometry and found that the levels of β1 and β3 were significantly reduced in Kind$^{Ko}$ and unaffected in Tln$^{Ko}$ cells (*Figure 2A* and *Figure 2—figure supplement 1*). The levels of α2 and α3 integrin were reduced in both cell lines, α6 was elevated in Tln$^{Ko}$ and decreased in Kind$^{Ko}$ cells, and the α3 levels were significantly more decreased in Kind$^{Ko}$ than in Tln$^{Ko}$ cells (*Figure 2A*) explaining the absent adhesion of both cell lines to COL and their differential adhesion behaviour on LN (*Figure 1D*). The β5 levels were similarly up-regulated in Kind$^{Ko}$ and Tln$^{Ko}$ cells, and the α5 and αv integrin levels were slightly reduced but not significantly different between Tln$^{Ko}$ and Kind$^{Ko}$ cells (*Figure 2A*). The differential adhesion of Mn$^{2+}$-treated Tln$^{Ko}$ and Kind$^{Ko}$ cells to VN (*Figure 1D*), despite similar surface levels of αv integrins, points to particularly important role(s) for kindlin-2 in αv integrins-VN adhesion and signaling (*Liao et al., 2015*). Serendipitously, the reduced expression of β1-associating α2, α3 and α6 subunits in Kind$^{Ko}$ cells, which impairs adhesion to LN and COL enables α5 to associate with the remaining β1 subunits and leads to comparable α5β1 levels on Tln$^{Ko}$ and Kind$^{Ko}$ cells (*Figure 2—figure supplement 2*) explaining their similar adhesion to FN (*Figure 1D,E* and *Figure 1—figure supplement 4*). Therefore, we performed all further experiments with FN.

Since we excluded different surface levels of FN-binding integrins as a cause for the severely compromised adhesion of Tln$^{Ko}$ and Kind$^{Ko}$ cells to FN, we tested whether talin and kindlin are required to activate FN-binding α5β1 integrins. To directly assess integrin activation, we made use of an antibody against the 9EG7 epitope, which specifically recognizes Mn$^{2+}$ and/or ligand activated β1 integrins (*Bazzoni et al., 1995*). The amount of 9EG7 epitope exposure relative to total β1 integrin exposure corresponds to the integrin activation index, which can be measured by flow cytometry using 9EG7 and anti-total β1 integrin antibodies. These measurements revealed that Tln$^{Ctr}$ and Kind$^{Ctr}$ cells bound 9EG7 antibodies, while Tln$^{Ko}$ and Kind$^{Ko}$ cells lacked 9EG7 binding in the absence of Mn$^{2+}$ (*Figure 2—figure supplement 3A*). Mn$^{2+}$ treatment significantly increased 9EG7 binding by Tln$^{Ctr}$ and Kind$^{Ctr}$ cells, which was further elevated in the presence of FN-Arg-Gly-Asp (RGD) ligand known to stabilize the high affinity state of integrins (*Figure 2—figure supplement 3A*). Mn$^{2+}$-treated Tln$^{Ko}$ and Kind$^{Ko}$ cells bound significantly less 9EG7 antibodies than control cells, which marginally increased with FN-RGD (*Figure 2—figure supplement 3A*). Moreover, the normalization of the 9EG7 binding to the total β1 integrin surface levels also indicated a significantly lower influence of Mn$^{2+}$ and FN-RGD on the integrin activation index of Kind$^{Ko}$ as compared to Tln$^{Ko}$ cells (*Figure 2—figure supplement 3A*). These findings confirm that both, talin and kindlin are required for β1 integrin activation and to stabilize Mn$^{2+}$-induced unbending/unclasping of α5β1 integrins.

Our findings so far suggest that talin and kindlin are required to activate FN-binding integrins and maintain Mn$^{2+}$-induced activation of FN-binding integrins. To further analyze whether ligand-induced stabilisation of high-affinity integrin conformations (also termed 'ligand-induced integrin activation; *Du et al., 1991*) can form in the absence of talin or kindlin, we used atomic force microscopy (AFM)-based single cell force spectroscopy (SCFS). We attached control, Tln$^{Ko}$ or Kind$^{Ko}$ cells to Concanavalin A (ConA)-coated cantilevers, allowed the cells to contact surfaces coated with either wild type FN-III$_{7\text{-}10}$ (FN-RGD) or an integrin-binding-deficient FN-III$_{7\text{-}10}$ fragment lacking the RGD binding motif (FN-ΔRGD) for increasing contact times, either in the absence or presence of Mn$^{2+}$ and then detached them from the substrate by lifting the cantilever (*Figure 2B,C*). In the absence of Mn$^{2+}$ Tln$^{Ctr}$ and Kind$^{Ctr}$ cells developed significant adhesion to FN-RGD within 5 s contact time. After a contact time of 20 s around 2 nN force was required to disrupt adhesion to FN-RGD, and after 50 and 120 s, respectively, 3 and 6 nN were required (*Figure 2B*). Tln$^{Ko}$ and Kind$^{Ko}$ cells failed to develop measurable adhesions to FN-RGD after contact times of 5, 20, 50 and 120 s (*Figure 2B*). Treatment with Mn$^{2+}$ induced a slight and similar increase of force required to disrupt adhesion of

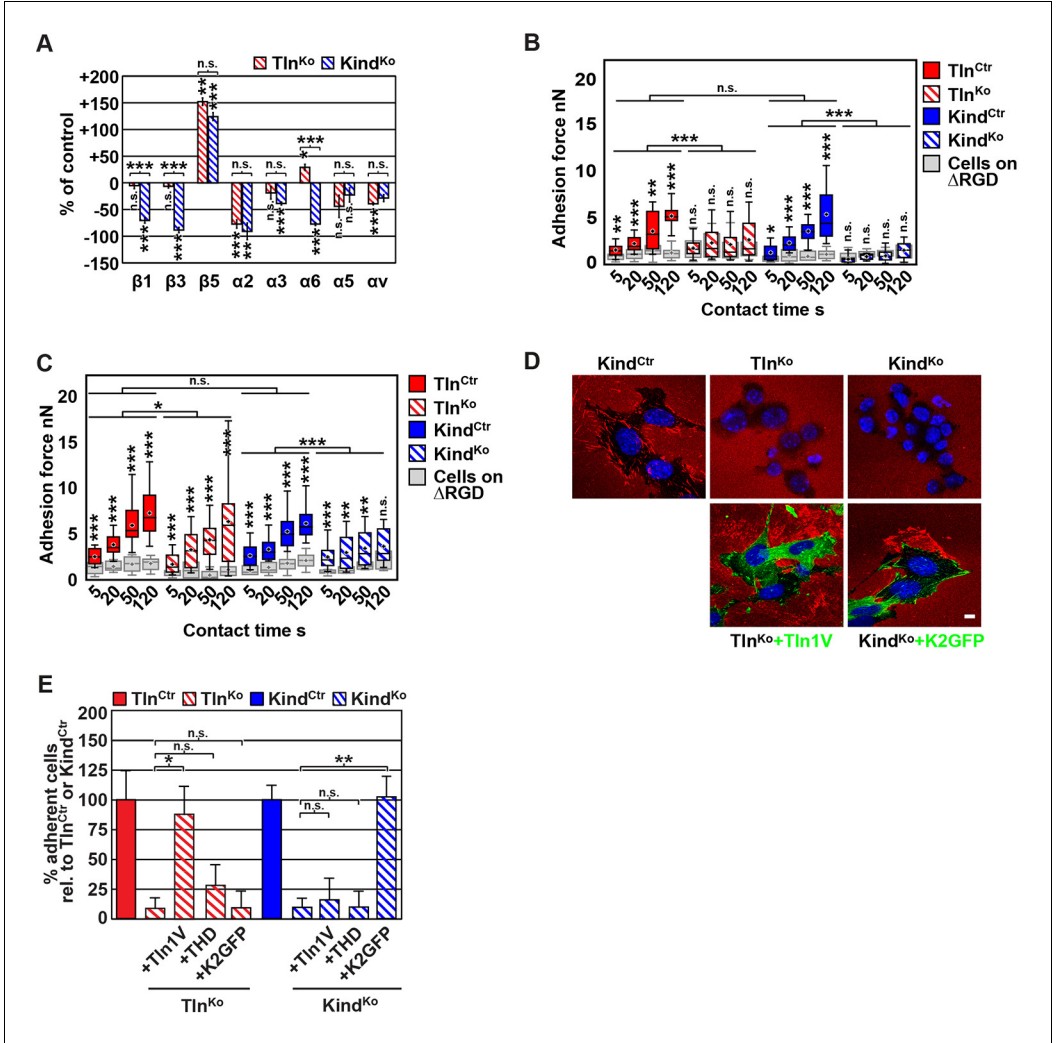

**Figure 2.** FN binding by Tln[Ko] and Kind[Ko] cells. (**A**) Quantification of integrin surface expression levels relative to the Tln[Ctr] and Kind[Ctr] cell lines; independent experiments: n=10 for β1; n=4 for β3, α5, αv; n=3 for remaining integrin subunits; error bars indicate standard error of the mean; significances are calculated between Tln[Ko] and Kind[Ko] cells indicated by brackets, or between Tln[Ko] or Kind[Ko] cells and corresponding control cells indicated by the significances above or below bars. (**B**, **C**) Box plot representation of adhesion forces generated by cells interacting with surface immobilized FN fragments. Cells were immobilized on ConA-coated AFM cantilevers and pressed onto surfaces coated with the FN-RGD or integrin-binding deficient FN-ΔRGD fragments for varying contact times, either in the absence (**B**) or presence of $Mn^{2+}$ (**C**). Coloured and grey boxplots represent adhesion forces from at least 10–15 independent experiments with single cells; + signs represent mean; the significance between adhesion on FN-RGD versus FN-ΔRGD is given on top of each boxplot and was calculated with a Mann–Whitney U test; brackets indicate two-way RM ANOVA comparisons of the whole adhesion kinetics. (**D**) FN staining after plating cells on a FN-coated dish for 24 hr. (**E**) Quantification of cell adhesion on FN 30 min after seeding; values are normalized to Tln[Ctr] and Kind[Ctr]; n=3 independent experiments; error bars indicate standard error of the mean. Bar, 10 μm. AFM, atomic force microscopy; ConA, Concanavalin A; FN, fibronectin; K2GFP, green fluorescent protein-tagged kindlin-2; RGD, Arg-Gly-Asp; RM ANOVA, repeated measures analysis of variance; THD, talin-1 head domain; Tln1V, Venus-tagged full length talin-1.

The following figure supplements are available for figure 2:

**Figure supplement 1.** Integrin expression profiles of Tln[Ctr], Tln[Ko], Kind[Ctr] and Kind[Ko] cells.

**Figure supplement 2.** Tln[Ko] and Kind[Ko] cells display comparable α5β1 integrin cell surface levels.

**Figure supplement 3.** β1 i ntegrin activation in Tln[Ctr], Tln[Ko], Kind[Ctr], Kind[Ko] cells.

**Figure supplement 4.** Re-expression of talin-1 or kindlin-2 in Tln[Ko] and Kind[Ko] cells.

control, $Tln^{Ko}$ and $Kind^{Ko}$ cells to FN-RGD after 5 s contact time (*Figure 2C*). However, with increasing contact times, the AFM profiles of $Tln^{Ko}$ and $Kind^{Ko}$ cells differ in the presence of $Mn^{2+}$. While the adhesion force increased concomitantly with longer contact times in $Tln^{Ctr}$, $Kind^{Ctr}$ and $Tln^{Ko}$ cells, adhesion forces of $Kind^{Ko}$ cells plateaued after 50 s and showed no further increase towards 120 s contact time. The latter finding suggests that kindlin stabilizes integrin–ligand complexes with time, by inducing integrin clustering and/or by modulating the off-rate of integrin ligand bonds, for example, through associating with the integrin-linked kinase (ILK)-Pinch-Parvin (IPP) complex that links kindlin to the F-actin cytoskeleton (*Cluzel et al., 2005*; *Ye et al., 2013*; *Montanez et al., 2008*; *Fukuda et al., 2014*).

We next tested whether their impaired integrin function affects the assembly of FN into fibrils, which requires association of active α5β1 integrin with the actin cytoskeleton (*Pankov et al., 2000*), and whether re-expression of talin and kindlin reverts the defects of $Tln^{Ko}$ and $Kind^{Ko}$ cells. While neither $Tln^{Ko}$ nor $Kind^{Ko}$ cells were able to assemble FN fibrils, re-expression of full-length Venus-tagged talin-1 (Tln1V) in $Tln^{Ko}$ or GFP-tagged kindlin-2 (K2GFP) in $Kind^{Ko}$ cells (*Figure 2—figure supplement 4*) rescued FN fibril assembly and adhesion to FN (*Figure 2D,E*). Furthermore, neither overexpression of the talin-1 head (THD) nor K2GFP in $Tln^{Ko}$ cells, nor Tln1V or THD in $Kind^{Ko}$ cells rescued adhesion to FN or 9EG7 binding (*Figure 2E* and *Figure 2—figure supplement 3B*).

Altogether, our results demonstrate that both talin and kindlin are required (1) for ligand-induced stabilisation of integrin-ligand complexes, (2) to stabilize $Mn^{2+}$-activated α5β1 integrins, and (3) to induce integrin-mediated FN fibril formation.

## $Tln^{Ko}$ cells initiate spreading and assemble β1 integrins at protruding membranes

It has been reported that a significant number of talin-2 small interfering RNA (siRNA)-expressing talin-1$^{-/-}$ fibroblasts adhere to FN and initiate isotropic cells spreading (*Zhang et al., 2008*). To test whether spreading can also be induced in adherent $Tln^{Ko}$ and $Kind^{Ko}$ cells, we bypassed their adhesion defect with $Mn^{2+}$, seeded them for 30 min on FN and stained with an antibody against total β1 integrin and the β1 integrin activation epitope-reporting 9EG7 antibody. As expected, $Tln^{Ctr}$ or $Kind^{Ctr}$ cells clustered 9EG7-positive β1 integrins in NAs and focal adhesions (FAs), whose frequency and size increased upon $Mn^{2+}$ treatment (*Figure 3A,B*). In contrast, the sporadic and very weakly adherent $Tln^{Ko}$ and $Kind^{Ko}$ cells were small, round and formed small and finely dispersed β1 integrin aggregates over the entire cell (*Figure 3A*) and lacked 9EG7-positive signals (*Figure 3B*) in the absence of $Mn^{2+}$ treatment. Upon $Mn^{2+}$ treatment 37 ± 1% (n=684, mean ± standard deviation of three independent experiments) of the $Tln^{Ko}$ cells showed isotropic membrane protrusions (circumferential lamellipodia) with small, dot-like aggregates of β1 integrin, kindlin-2, paxillin and ILK at the membrane periphery (*Figure 3A* and *Figure 3—figure supplement 1*), which eventually detached from the substrate leading to the collapse of the protruded membrane (*Video 1*). Furthermore, 9EG7-positive β1 integrins accumulated along the lamellipodial edge and beneath the nucleus of $Tln^{Ko}$ cells (*Figure 3B*). The remaining cells were spheroid, with half of them showing short, finger-like protrusions, which were motile due to their poor anchorage to the substrate. In the case of $Kind^{Ko}$ cells, we analysed 652 cells in three independent experiments and found that only 7 ± 1% (mean ± standard deviation) of the cells established lamellipodia, which formed around the entire circumference in 2 ± 0.4% (mean ± standard deviation) of the cells. Around 93 ± 1% of the $Kind^{Ko}$ cells were spheroid (mean ± standard deviation) and frequently had finger-like, motile protrusions with small dot-like signals containing β1 integrin and talin but rarely paxillin or ILK (*Figure 3A* and *Figure 3—figure supplement 1*). Importantly, re-expression of Tln1V in $Tln^{Ko}$ cells or K2GFP in $Kind^{Ko}$ cells normalized FA formation and spreading on FN (*Figure 3—figure supplement 2*). These findings indicate that kindlin-2 expressing $Tln^{Ko}$ cells can initiate the formation of large lamellipodia and assemble β1 integrins in lamellipodial edges.

To further characterize the distribution of β1 integrins in the lamellipodial edges of $Tln^{Ko}$ cells, we visualized them by combining direct stochastic optical reconstruction microscopy (dSTORM) and total internal reflection fluorescence microscopy (TIRFM). $Mn^{2+}$-treated and non-permeablized cells were seeded on FN, stained with anti-total β1 integrin antibodies, and then permeabilized, immunostained for paxillin and imaged with normal resolution TIRFM and dSTORM (*Figure 3C*). Each localization detected by dSTORM was plotted as a Gaussian distribution around its centre with an average spatial accuracy of ~20 nm (resolution limit of dSTORM imaging). Since two or more

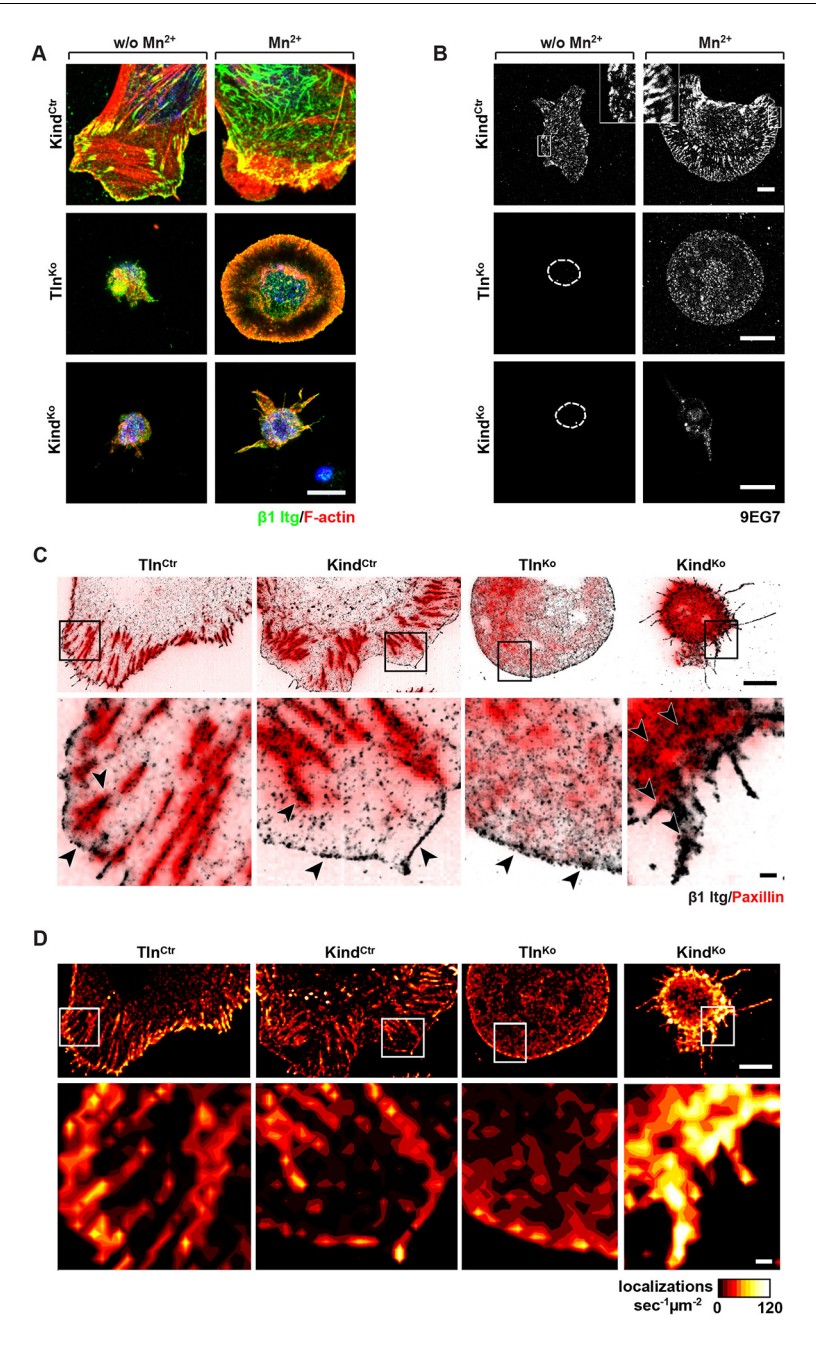

**Figure 3.** Integrin distribution in Tln[Ko] and Kind[Ko] cells. (**A**) Confocal images of the ventral side of adherent cells stained for β1 integrin and F-actin in the absence or presence of Mn²⁺ stimulation. Notice the increase in the spreading area (w/o Mn²⁺: 1696 ± 360 μm², Mn²⁺: 2676 ± 466 μm²) and in the average size (w/o Mn²⁺: 0.64 ± 0.1 μm², Mn²⁺: 0.89 ± 0.1 μm²) and number (w/o Mn²⁺: 105 ± 38, Mn²⁺: 246 ± 8) of focal adhesions in Kind[Ctr] cells after Mn²⁺ stimulation and the increase of spreading area in the Tln[Ko] (w/o Mn²⁺: 77 ± 1 μm², Mn²⁺: 572 ± 37 μm²) and Kind[Ko] cells (w/o Mn²⁺: 76 ± 27 μm², Mn²⁺: 152 ± 8 μm²) (n=3, mean ± standard deviation). (**B**) Confocal images from the ventral side of adherent cells stained for the 9EG7 epitope in the absence or presence of Mn²⁺ stimulation. (**C**) TIRF-dSTORM images of β1 integrin (grey scale image) obtained from immunostaining of non-permeabilized cells overlaid with anti-paxillin staining following permeabilization (red, normal resolution). Boxed areas are displayed in a five-fold magnification. (**D**) Images show heat map representations of dSTORM localizations per μm² and sec, indicative for integrin clustering defined by local integrin densities. The colour range indicates localizations s⁻¹ μm⁻² with low values shown in dark red colours and high densities from yellow to white colours. Bars, 10 μm (A,B); 5 μm (C,D); 500 nm (for the magnification in C,D). TIRF, total internal reflection fluourescence; dSTORM, direct stochastic optical reconstruction microscopy.

The following figure supplements are available for figure 3:

*Figure 3 continued on next page*

**Figure supplement 1.** Localization of FAs proteins in Mn$^{2+}$-treated Kind$^{Ctr}$, Tln$^{Ko}$ and Kind$^{Ko}$ cells.
DOI: 10.7554/eLife.10130.015

**Figure supplement 2.** Rescue of FA formation and spreading after expression of Tln1V in Tln$^{Ko}$ cells or K2GFP in Kind$^{Ko}$ cells.
DOI: 10.7554/eLife.10130.016

**Figure supplement 3.** Distribution of β1 integrins in spheroid-shaped Tln$^{Ko}$ cells.
DOI: 10.7554/eLife.10130.017

localizations from single or multiple dyes in close proximity cannot be distinguished, the number of localizations does not directly reflect integrin numbers. However, all antibody molecules display the same average behaviour with respect to the number of localizations per second in all areas of the cell. This allowed to average the number of localizations per second and µm$^2$ and to plot them in a heat map representation (*Figure 3D*), which directly reflects the density of stained β1 integrin molecules and thus the degree of integrin clustering. The β1 integrin staining of Tln$^{Ctr}$ and Kind$^{Ctr}$ cells revealed small round structures of ~50 nm diameter indicating clusters of integrins larger than the resolution limit (*Figure 3C*; high magnification; see arrowheads). Furthermore, high numbers of localizations were enriched in paxillin-positive FAs and in NAs at the lamellipodial edge (*Figure 3C*; see arrowheads). In these areas, we observed a high average density of 60–120 localizations s$^{-1}$ µm$^{-2}$, while outside of the adhesion sites ~0–20 localizations s$^{-1}$ µm$^{-2}$ were detected, indicating a high degree of β1 integrin clustering within and a low degree of clustering outside of adhesion sites (*Figure 3D*). Tln$^{Ko}$ cells with circumferential lamellipodia showed a high density of blinking with up to 100 localizations s$^{-1}$ µm$^{-2}$ at lamellipodial edges (*Figure 3C,D*; see arrowheads), which appeared less compact than in control cells. Kind$^{Ko}$ cells showed >120 localizations s$^{-1}$ µm$^{-2}$ in the periphery and finger-like membrane protrusions (*Figure 3C,D*; see arrowheads), which were also observed in Tln$^{Ko}$ cells that adopted a spheroid rather than an isotropic spread shape (*Figure 3—figure supplement 3*). The exclusive presence of these large and entangled β1 integrin aggregates on Tln$^{Ko}$ and Kind$^{Ko}$ cells with small, spheroid shapes and protrusions suggests that they were induced by spatial constraints rather than specific signaling.

These findings demonstrate that, in contrast to Kind$^{Ko}$ cells, Mn$^{2+}$-treated kindlin-2-expressing Tln$^{Ko}$ cells induce circumferential membrane protrusions with β1 integrins at the protrusive edges.

## Kindlin-2 binds and recruits paxillin to NAs

Our data so far indicate that the expression of kindlin-2 enables initial, isotropic spreading and the accumulation of integrins in lamellipodia of Mn$^{2+}$-treated Tln$^{Ko}$ cells. To identify binding partner(s) of kindlin-2 that transduce this function to downstream effectors, we performed yeast-two-hybrid assays with kindlin-2 as bait using a human complementary DNA (cDNA) library containing all possible open reading frames and a human keratinocyte-derived cDNA library. Among the 124 cDNAs identified from both screenings, 17 coded for leupaxin and 11 for Hic-5. Immuno-precipitation of overexpressed green fluorescent protein (GFP)-tagged paxillin family members, paxillin, Hic-5 and leupaxin in HEK-293 cells with an anti-GFP antibody efficiently co-precipitated FLAG-tagged kindlin-2 (K2flag) (*Figure 4A*). Conversely, overexpressed GFP-tagged kindlin family members (kindlin-1, kindlin-2, kindlin-3) co-precipitated Cherry-paxillin (*Figure 4—figure supplement 1*). Since fibroblasts express high levels of paxillin (*Figure 4—figure supplement 2*), we performed all further interaction analysis with paxillin. Immunoprecipitations of GFP-tagged paxillin or kindlin-2 truncation mutants (*Figure 4—figure supplement 3A*) revealed that

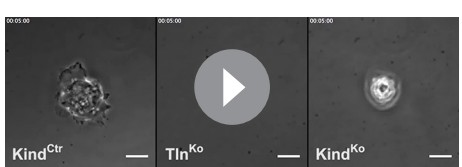

**Video 1.** Spreading Kind$^{Ctr}$, Tln$^{Ko}$ and Kind$^{Ko}$ cells on FN. Assembled time lapse movies of Kind$^{Ctr}$, Tln$^{Ko}$ and Kind$^{Ko}$ cells. Cell spreading was recorded 5 min after seeding on FN. Kind$^{Ctr}$ cells were already well spread and only a minor size increase was observed over the following minutes. The Tln$^{Ko}$ cells formed a circumferential lamellipodium that rapidly collapsed and subsequently the cells formed finger-like protrusions of varying size and failed to reestablish a fully formed circular lamellipodium. The Kind$^{Ko}$ cells failed to form a lamellipodium and formed finger-like protrusions that were not always adherent. Bar, 10 µm. FN, fibronectin.

the interaction between kindlin-2 and paxillin was dramatically reduced in the absence of the Lin-11, Isl-1 and Mec-3 (LIM)1-4, LIM2-4 or LIM3-4 domains of paxillin (*Figure 4B*), or the pleckstrin homology (PH) domain (K2ΔPHGFP; lacking amino acids 380-477) or the N-terminus of kindlin-2 including the F0, F1, and the N-terminal part of the F2 domains (K2NTGFP; terminating at the end of F1; spanning amino acids 1-229) (*Figure 4C*). As expected, the interaction between kindlin-2 and ILK (*Montanez et al., 2008*), which is mediated via a recently identified sequence in the linker domain between the end of the N-terminal F2 and the beginning of the PH domain (amino acids 353–357) (*Fukuda et al., 2014*; *Huet-Calderwood et al., 2014*), was abolished by the K2NTGFP truncation but unaffected by the deletion of the PH domain (K2ΔPHGFP) or the deletion of the N-terminal F0 and F1 domains (K2CTGFP, spanning amino acids 244-680) (*Figure 4C*). Importantly, immunoprecipitation of Kind[Ctr] lysates with antibodies against paxillin co-precipitated kindlin-2 (*Figure 4D*), confirming interactions between the endogenous proteins. Pull down experiments with recombinant full-length paxillin or paxillin-LIM3 domain and recombinant kindlin-2 demonstrated that binding to LIM3 and full-length paxillin is direct, $Zn^{2+}$-dependent and abrogated with ethylenediaminetetraacetic acid (EDTA) (*Figure 4E* and *Figure 4—figure supplement 3B*). Kind[Ko] cells were transduced with K2GFP or K2ΔPHGFP expression constructs, seeded on FN for different times and stained for β1 integrin, paxillin and F-actin. The experiments revealed that the expression of K2GFP in Kind[Ko] cells rescued spreading and induced robust paxillin recruitment to β1 integrin-positive NAs (*Figure 4F, G*). In contrast, expression of K2ΔPHGFP failed to recruit paxillin to β1 integrin-positive adhesion sites at the rim of membrane protrusions (*Figure 4F,G*) and induce normal cell spreading (*Figure 4—figure supplement 4A*) despite proper, although weaker, localisation to β1 integrin-positive adhesion sites (*Figure 4—figure supplement 4B,C*). Interestingly, mature FAs in K2ΔPHGFP-expressing cells were prominent after 30 min and contained significant amounts of paxillin, indicating that paxillin is recruited to mature FAs in a kindlin-2-independent manner (*Figure 4F*).

These findings indicate that the PH domain of kindlin-2 directly binds the LIM3 domain of paxillin and recruits paxillin into NAs but not into mature FAs.

## The kindlin-2/paxillin complex promotes FAK-mediated cell spreading

Our findings revealed that kindlin-2 is required to recruit paxillin to NAs. Paxillin in turn, was shown to bind, cluster and activate FAK in NAs, which leads to the recruitment of p130Cas, Crk and Dock followed by the activation of Rac1 and the induction of cell spreading, and, in concert with growth factor signals, to the activation of Akt-1 followed by the induction of cell proliferation and survival (*Schlaepfer et al., 2004*; *Bouchard et al., 2007*; *Zhang et al., 2014*; *Brami-Cherrier et al., 2014*). We therefore hypothesized that the recruitment of paxillin and FAK by kindlin-2 triggers the isotropic spreading and expansion of Tln[Ko] cells. To test this hypothesis, we seeded our cell lines on FN or poly-L-lysine (PLL) in the presence or absence of epidermal growth factor (EGF) and $Mn^{2+}$. We found that EGF induced similar phosphorylation of tyrosine-992 (Y992) of the epidermal growth factor receptor (pY992-EGFR) in control, Tln[Ko] and Kind[Ko] cells. The phosphorylation of tyrosine-397 of FAK (pY379-FAK) in Kind[Ctr] cells was strongly induced after the adhesion of control cells on FN and was not further elevated after the addition of EGF and $Mn^{2+}$ (*Figure 5A* and *Figure 5—figure supplement 1*). Tln[Ko] cells also increased pY397-FAK as well as pY31-Pxn and pY118-Pxn levels upon adhesion to FN, however, significantly less compared to control cells (*Figure 5A* and *Figure 5—figure supplement 1A-C*). Furthermore, EGF and $Mn^{2+}$ treatments further increased pY397-FAK levels in Tln[Ko] cells and localized pY397-FAK to peripheral NA-like adhesions (*Figure 5A,B* and *Figure 5—figure supplement 1A-C*). In sharp contrast, Kind[Ko] cells seeded on FN or treated with EGF and $Mn^{2+}$ failed to induce pY397-FAK, pY31-Pxn, pY118-Pxn (*Figure 5A* and *Figure 5—figure supplement 1A-C*) and localize pY397-FAK to peripheral membrane regions (*Figure 5B*). Importantly, re-expression of Talin1-Venus in Tln[Ko] and Kindlin2-GFP and Kind[Ko] cells fully rescued these signaling defects (*Figure 5—figure supplement 1B,C*). Furthermore, stable expression of K2GFP in Kind[Ko] cells rescued pY397-FAK and pS473-Akt levels (*Figure 5C*) and co-precipitated paxillin and FAK with K2GFP (*Figure 5—figure supplement 2*). In contrast, stable expression of K2ΔPHGFP in Kind[Ko] cells failed to co-precipitate paxillin and FAK (*Figure 5—figure supplement 2*) and induce pY397-FAK and pS473-Akt (*Figure 5C*).

In line with previous reports showing that the paxillin/FAK complex can trigger the activation of p130Cas (*Zhang et al., 2014*) and, in cooperation with EGFR signaling, the activation of Akt (*Sulzmaier et al., 2014*, *Deakin et al., 2012*), we observed Y410-p130Cas, pT308-Akt, S473-Akt and

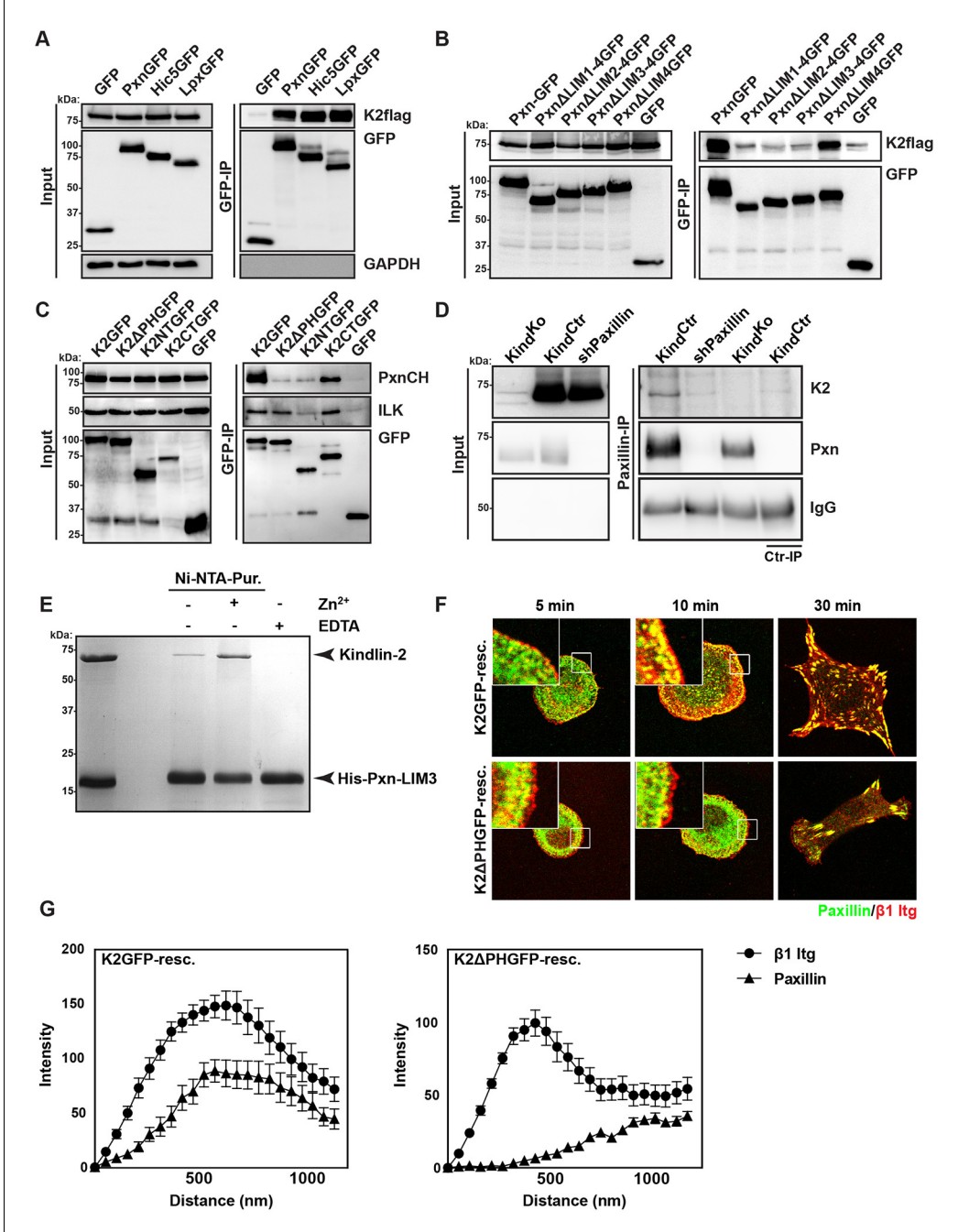

**Figure 4.** Kindlin binds and recruits paxillin to NAs. (**A**) GFP-IP of lysates from HEK 293T cells overexpressing GFP-tagged paxillin, Hic5 and leupaxin constructs (Pxn, paxillin; Hic5; Lpx, leupaxin) and K2flag reveal interaction of kindlin-2 with all three paxillin family members. (**B**) GFP-IP of lysates from HEK 293T cells overexpressing GFP-tagged paxillin truncation mutants and K2flag identifies the paxillin LIM3 domain as kindlin-2-binding domain. (**C**) GFP-IP of lysates from HEK 293T cells overexpressing GFP-tagged kindlin-2 truncation/deletion mutants and Cherry-tagged paxillin (PxnCH) identifies the kindlin-2 PH domain as paxillin binding domain. (**D**) Co-IP of endogenous paxillin and kindlin-2 from Kind[Ctr] cells. (**E**) Purified His-tagged paxillin-LIM3 domain pulls down recombinant kindlin-2 in a Zn[2+]-dependent manner. (**F**) K2GFP and K2ΔPHGFP expressing Kind[Ko] cells seeded on FN for the indicated times and stained for paxillin and β1 integrin. (**G**) Fluorescence intensity line scans from K2GFP- (n=11 cells) and K2ΔPHGFP- (n=17 cells) expressing Kind[Ko] cells cultured on FN for 10 min and stained for paxillin and β1 integrin; error bars indicate standard error of the mean. Bar, 10 μm. EDTA, ethylenediaminetetraacetic acid; FN, fibronection; GAPDH, glyceraldehyde-3-phosphate dehydrogenase; GFP, green fluorescent protein; ILK, integrin-linked kinase; IP, immunoprecipitation; K2GFP, green fluorescent protein-tagged kindlin-2; LIM, Lin-11, Isl-1 and Mec-3; NAs, nascent adhesions; PH, pleckstrin homology.

*Figure 4 continued on next page*

*Figure 4 continued*

The following figure supplements are available for figure 4:

**Figure supplement 1.** Kindlin-1, -2 and -3 interact with paxillin.

**Figure supplement 2.** Expression of paxillin family members in different cell lines.

**Figure supplement 3.** Direct interaction between paxillin and kindlin-2.

**Figure supplement 4.** K2ΔPHGFP fails to recruit paxillin to β1 integrin-positive adhesions in Kind[Ko] cells.

pT202/pY204 Erk1/2 phosphorylation after $Mn^{2+}$ and/or EGF treatment of FN-seeded control and rescued cells, and to a slightly lesser extent Tln[Ko] cells (*Figure 5D*, *Figure 5— figure supplement 3A,B*). In contrast, FN-seeded Kind[Ko] cells failed to activate p130Cas and showed reduced Akt and Erk1/2 phosphorylation in response to EGF (*Figure 5D*, *Figure 5—figure supplement 3A,B*).

Finally, we tested whether the impaired activity of FAK contributed to the spreading defect of Kind[Ko] cells by chemically inhibiting FAK activity in Tln[Ko] cells or by overexpressing FAK in Kind[Ko] cells (*Figure 5E–G*). The experiments revealed that inhibiting FAK reduced lamellipodia formation of Tln[Ko] cells to an extent that was similar to untreated Kind[Ko] cells (*Figure 5E*). Conversely, overexpression of FAKGFP in Kind[Ko] cells resulted in high active FAK, increased lamellipodial formation and increased cell spreading in Tln[Ko] and Kind[Ko] cells (*Figure 5F,G* and *Figure 5—figure supplement 4A,B*).

Altogether, these findings show that the kindlin-2/paxillin complex in NAs recruits and activates FAK to induce cell spreading and increase the strength of Akt signaling.

## Discussion

While the functions of talin and kindlin for integrin activation, adhesion and integrin-dependent signaling in hematopoietic cells are firmly established, their roles for these processes in non-hematopoietic cells are less clear. To clarify this issue, we established mouse fibroblast cell lines that lacked either talin-1/2 (Tln[Ko]) or kindlin-1/2 (Kind[Ko]) and tested whether they were able to activate integrins and mediate substrate adhesion and signaling. In line with previous reports (*Bottcher et al., 2012*; *Margadant et al., 2012*), the deletion of *Tln1/2* or *Fermt1/2* genes changed the surface levels of laminin- and collagen-binding integrins. Since surface levels of α5 and αv integrins remained unchanged between Tln[Ko] and Kind[Ko] cells, we were able to establish the specific roles of talin and kindlin for the function of FN-binding integrins under identical conditions.

A major finding of our study demonstrates that integrin affinity regulation (activation) is essential for fibroblast adhesion and depends on both talin and kindlin-2 (*Figure 6A,D*). The unambiguity of this finding was unexpected in light of several reports showing that integrin activation and integrin-mediated adhesion still occurs in talin-depleted cells, or is inhibited when kindlin-2 is overexpressed (*Harburger et al., 2009*; *Wang et al., 2011*; *Lawson et al., 2012*). The previous studies that addressed the functional properties of talin used siRNA-mediated protein depletion, a combination of gene ablation and siRNA technology, or approaches to interfere with talin recruitment to NAs either by ablating the talin upstream protein FAK or by expressing an integrin that harbors a mutation in the talin binding site. Since the majority of approaches deplete rather than eliminate proteins from cells and adhesion sites, the respective cells were most likely recruiting sufficient residual protein to adhesion sites to allow integrin activation, cell adhesion and spreading, and the assembly of adhesion- and signaling-competent NAs. It is possible that not all integrin molecules have to be occupied by talin and therefore low levels of talin suffice, particularly in NAs that were shown by fluorescence correlation spectroscopy to contain only half the number of talin relative to α5β1 integrin and kindlin-2 molecules (*Bachir et al., 2014*). However, when the entire pool of talin is lost or decreased below certain thresholds (*Margadant et al., 2012*) integrins remain inactivate and consequently adhesion sites do not form. With respect to kindlin, it was reported that overexpressed kindlin-2 in CHO cells inhibits rather than promotes talin head domain-induced α5β1 integrin activation (*Harburger et al., 2009*). An integrin inhibiting effect of kindlin-2 is inconsistent with our study,

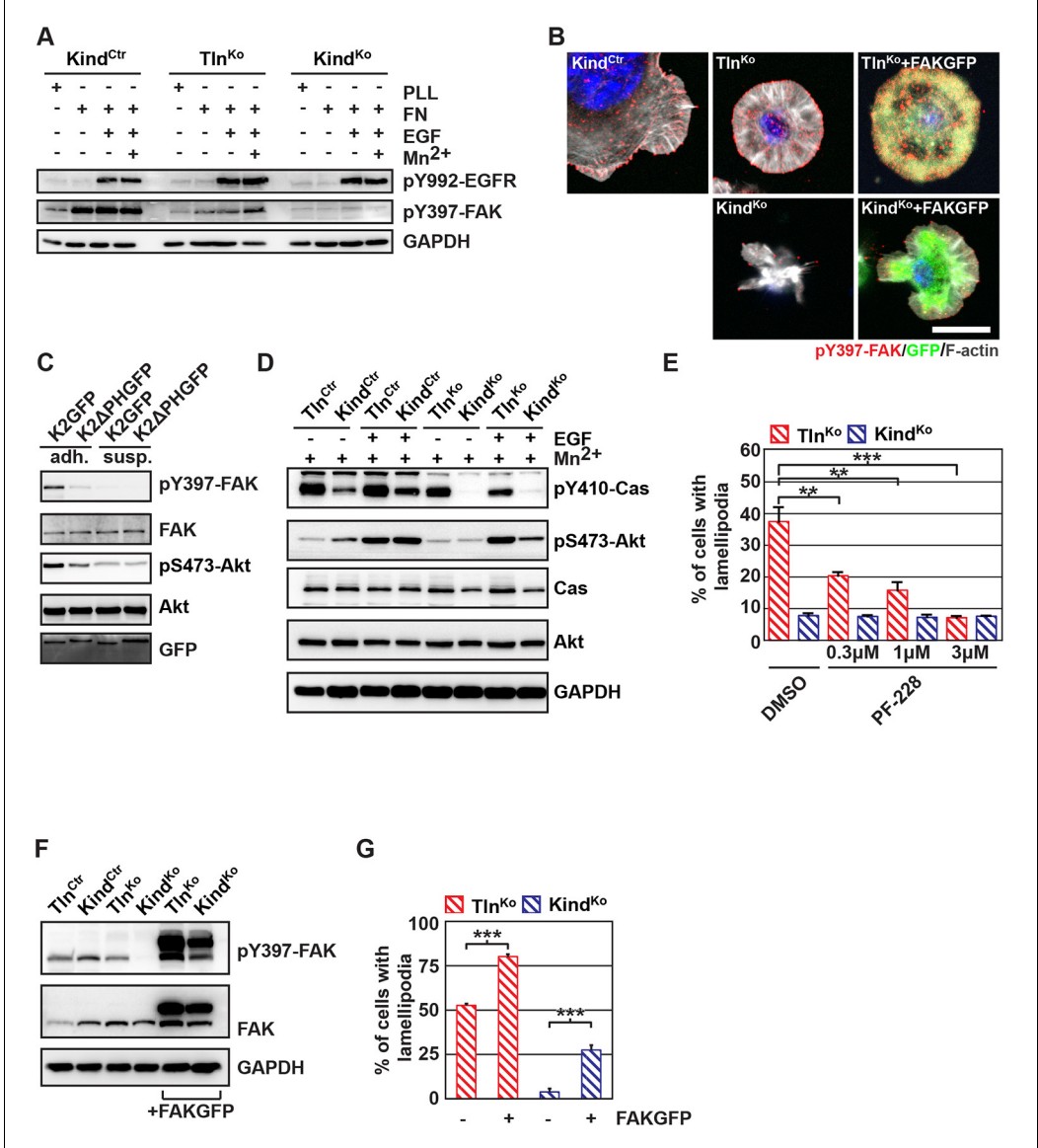

**Figure 5.** The kindlin/paxillin complex induces FAK signaling and cell spreading. (**A**) FAK and EGFR activation after seeding serum-starved Kind$^{Ctr}$, Tln$^{Ko}$ and Kind$^{Ko}$ cells on PLL or FN and treating them with or without EGF and Mn$^{2+}$. (**B**) Immunofluorescence staining of activated (Tyr-397 phosphorylated) FAK and F-actin in cells seeded on FN and treated with Mn$^{2+}$ for 30 min (FAKGFP indicates exogenous expression of FAKGFP fusion protein). (**C**) FAK and Akt activation in Kind$^{Ko}$ cells stably transduced with K2GFP or K2ΔPHGFP either seeded on FN or kept in suspension. GFP indicates similar expression of transduced GFP-tagged constructs. GAPDH levels served to control loading. (**D**) Levels of phosphorylated signaling mediators downstream of FAK in Mn$^{2+}$-treated, serum-starved or EGF-treated Kind$^{Ctr}$, Tln$^{Ko}$ and Kind$^{Ko}$ cells. GAPDH levels served to control loading. (**E**) Quantification of lamellipodia formation of FN-seeded Tln$^{Ko}$ and Kind$^{Ko}$ cells treated with Mn$^{2+}$ and either DMSO or the FAK inhibitor PF-228 (n=3 independent repeats; >100 cells/condition; error bars indicate standard error of the mean; significances are given in comparison to the DMSO control). (**F**) FAK activity in Tln$^{Ko}$ and Kind$^{Ko}$ cells stably transduced with FAKGFP (n=3 independent experiments). (**G**) Quantification of lamellipodia formation in Tln$^{Ko}$ and Kind$^{Ko}$ cells stably transduced with FAKGFP (n=3 independent experiments; significances are given in comparison to untreated control; error bars indicate standard error of the mean). Bar, 10 μm. DMSO, dimethyl sulfoxide; EGF, epidermal growth factor; EGFR, epidermal growth factor receptor; FAK, focal adhesion kinase; FAKGFP, green fluorescent protein-tagged FAK; FN, fibronectin; GAPDH, glyceraldehyde-3-phosphate dehydrogenase; GFP, green fluorescent protein; PLL, poly-L-lysine.

The following figure supplements are available for figure 5:

**Figure supplement 1.** FAK phosphorylation in Tln$^{Ctr}$, Tln$^{Ko}$, Tln$^{Ko+T1V}$, Kind$^{Ctr}$, Kind$^{Ko}$ and Kind$^{Ko+K2GFP}$ cells.

**Figure supplement 2.** Kindlin-2 forms a ternary complex with paxillin and FAK.

*Figure 5 continued on next page*

*Figure 5 continued*

**Figure supplement 3.** Activity of signaling mediators downstream of FAK in Tln^Ctr, Tln^Ko, Tln^Ko+T1V, Kind^Ctr, Kind^Ko and Kind^Ko+K2GFP cells.

**Figure supplement 4.** Cell spreading of FAK overexpressing Tln^Ko and Kind^Ko cells.

which identified a crucial role for kindlin in integrin activation, as well as with other studies also demonstrating that kindlin-2 promotes integrin functions (*Montanez et al., 2008*). It could well be that the reported inhibition of α5β1 by kindlin-2 represents an artifact that arose from protein overexpression.

Integrin activation can be induced with $Mn^{2+}$, whose binding to the ectodomain of β subunits directly shifts integrins into the high affinity state without the requirement for inside-out signals (*Mould et al., 1995*). We observed that $Mn^{2+}$-treated Tln^Ko and Kind^Ko cells expressed the activation-dependent epitope 9EG7 and adhered to FN, albeit at significantly lower levels and efficiencies than the normal parental or rescued cells (*Figure 6B,C*). This observation strongly indicates that talin and kindlin also cooperate to maintain the extended and unclasped conformation of active integrins. Although it is not known how talin and kindlin keep integrins in an active state, it is possible that they stabilize this conformation by linking the unclasped β integrin cytoplasmic domain to the plasma membrane and/or to cortical actin, which may firmly hold separated integrin α/β subunits apart from each other. The expression of mutant talins and kindlins in our cells should allow us to examine these possibilities in future.

Finally, our study also revealed that $Mn^{2+}$-treated Tln^Ko cells began to form large, circumferential lamellipodia that eventually detached from FN, leading to the collapse of the protruded membrane. This initial isotropic spreading was significantly less frequent in Kind^Ko cells, and has also been observed in talin-2-depleted talin-1^−/− cells on FN, although these cells did not require $Mn^{2+}$ for inducing spreading, which is likely due to the presence of residual talin-2 that escaped siRNA-mediated depletion (*Zhang et al., 2008*; *Zhang et al., 2014*). These findings strongly suggest that integrin binding to FN enables kindlin-2 in Tln^Ko cells to cluster β1 integrins (as shown for αIIbβ3 by kindlin-3 in *Ye et al., 2013*) and to trigger a signaling process that initiates spreading.

To find a mechanistic explanation for the kindlin-2-mediated cell spreading, we used the yeast-two-hybrid technology to identify paxillin as a novel and direct binding partner of kindlin-2. The interaction of the two proteins occurs through the LIM3 domain of paxillin, which was previously identified as integrin adhesion-targeting site (*Brown et al., 1996*), and the PH domain of kindlin-2. It is not unusual that PH domains fulfill dual roles by binding phospholipids and proteins, either simultaneously or consecutively (*Scheffzek and Welti, 2012*). The expression of a PH domain-deficient kindlin-2 in Kind^Ko cells rescues adhesion to FN and FA maturation, however, significantly impairs spreading and plasma membrane protrusions. This finding together with the observations that paxillin-null fibroblasts and embryonic stem cells have defects in spreading, adhesion site remodeling and formation of lamellipodia (*Hagel et al., 2002*, *Wade et al., 2002*) indicates that the kindlin-2/paxillin complex induces the elusive signaling process, leading to initial spreading of Tln^Ko and talin-depleted cells (*Zhang et al., 2008*). Indeed, the kindlin-2/paxillin complex in NAs recruits FAK (*Deramaudt et al., 2014*, *Thwaites et al., 2014*; *Choi et al., 2011*), which cooperates with growth factor receptors (such as EGFR) to induce signaling pathways that activate Erk and Akt to promote proliferation and survival, as well as Arp2/3 and Rac1 to induce actin polymerization and membrane protrusions (*Figure 6B,E*). Kindlin-2 also recruits ILK, which binds in the vicinity of the kindlin-2 PH domain and links integrins to actin and additional signaling pathways (*Figure 6E*). The short-lived nature of the initial spreading of Tln^Ko and talin-depleted (*Zhang et al., 2008*) cells shows that talin concludes the integrin-mediated adhesion process in NAs (*Figure 6F*) and induces the maturation of FAs. The formation of paxillin-positive FAs in cells expressing the PH domain-deficient kindlin-2 suggests that the recruitment of paxillin to FAs occurs either in a kindlin-independent manner or through a modification of kindlin in a second binding motif.

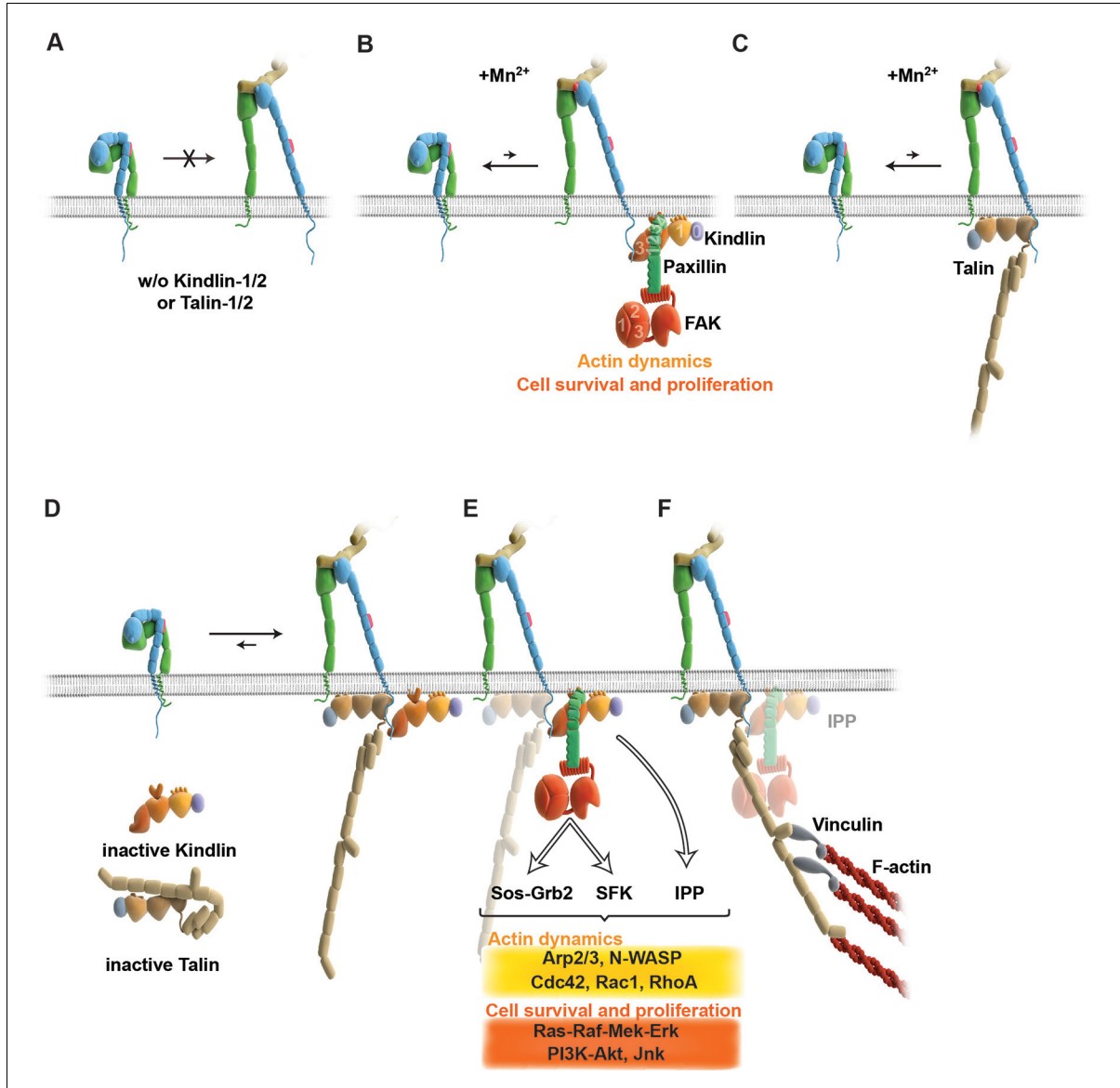

**Figure 6.** Model for the roles of talin and kindlin during inside-out and outside-in signaling of α5β1 integrin. Integrin subunits are modelled according to *Zhu et al. (2008)*, with the α5 subunit in green and the β1 subunit in blue showing the bent and clasped low affinity and the extended and unclasped high affinity conformations; the 9EG7 epitope is marked as red dot at the β1 leg and the FN ligand as beige dimers. (**A**) α5β1 integrin fails to shift from a bent to an extended/unclasped, high affinity state in the absence of talin-1/2 or kindlin-1/2; the bent/clasped conformation brings the EGF-2 domain of the β subunit in close contact with the calf domain of the α5 subunit and prevents exposure of the 9EG7 epitope. (**B**) In the absence of talin (Tln^Ko) and presence of $Mn^{2+}$, kindlin-2 allows adhesion by stabilizing the high affinity conformation of a low number of integrins and the direct binding of paxillin, leading to nucleation of integrins, recruitment of FAK, FAK-dependent signaling and lamellipodia formation. (**C**) In the absence of kindlins (Kind^Ko), talin stabilizes the high affinity conformation of a low number of integrins but does not enable paxillin recruitment and lamellipodia formation. (**D**) In normal fibroblasts, binding of kindlin and talin to the β1 tail is associated with the stabilisation of the unclasped α5β1 heterodimer and 9EG7 epitope exposure. (**E**) Kindlin recruits paxillin and FAK through the kindlin-PH domain and ILK/Pinch/Parvin (IPP; not shown) in a talin-independent manner and induces cell spreading, proliferation and survival. (**F**) The high affinity conformation of α5β1 integrin is stabilized by linkage of the β1 tail to the actin cytoskeleton through talin (and potentially the IPP complex; not shown). The arrow length indicates integrin conformations existing at equilibrium. EGF, epidermal growth factor; FAK, focal adhesion kinase; FN, fibronectin; ILK, integrin-linked kinase; IPP, integrin-linked kinase-Pinch-Parvin; SFK, src family kinases.

## Materials and methods

### Mouse strains and cell lines and cell culture

The floxed kindlin-1 (*Fermt1$^{flox/flox}$*), floxed talin-1 (*Tln1$^{flox/flox}$*) and the constitutive talin-2-null (*Tln2$^{-/-}$*) mouse strains have been described (*Rognoni et al., 2014*; *Nieswandt et al., 2007*; *Conti et al., 2009*). The floxed kindlin-2 (*Fermt2$^{flox/flox}$*) mouse strain generated via recombinant recombination in embryonic stem cells (*Fassler and Meyer, 1995*) carries *loxP* sites flanking exon 15, which contains the stop codon and the polyadenylation signal of the murine *Fermt2* gene. Homologous recombination and germ line transmission were verified by Southern blots, and the *frt*-flanked neo cassette was removed with a transgenic mouse strain carrying a *deleter-flipase* gene. Floxed talin-1 and talin-2-null mice, and floxed kindlin-1 and kindlin-2 mice were intercrossed to generate *Tln1$^{flox/flox}$ Tln2$^{-/-}$* and *Fermt1$^{flox/flox}$ Fermt2$^{flox/flox}$* mice.

The cell lines used in this study are mouse fibroblasts derived from the kidneys of 21 d old mice, immortalized by retrovirally transducing the SV40 large T antigen, cloned (Tln$^{Ctr}$ and Kind$^{Ctr}$) and finally infected with an adenovirus to transduce the *Cre* recombinase resulting in talin-null (Tln$^{Ko}$) and kindlin-null (Kind$^{Ko}$) cells. The parental cell lines were authenticated based on morphological criteria and the surface expression of specific integrins. All cells were cultured under standard cell culture conditions using Dulbecco's modified Eagle's medium (DMEM) supplemented with 8% fetal calf serum (FCS) and Penicillin/Streptomycin but not subjected to mycoplasma contamination testing.

### Flow cytometry

Flow cytometry was carried out with a FACSCantoTMII cytometer (BD Biosciences, Franklin Lakes, NJ, USA) equipped with FACS DiVa software (BD Biosciences) using standard procedures. Data analysis was carried out with the FlowJo program (version 9.4.10). Fibroblasts were incubated with primary antibodies diluted in FACS-Tris buffered saline (FACS-TBS; 30 mM Tris, pH 7.4, 180 mM NaCl, 3.5 mM KCl, supplemented with 1 mM CaCl$_2$, 1 mM MgCl$_2$, 3% BSA, 0,02% NaN$_3$) for 1 hr on ice, washed twice with cold FACS-TBS and finally incubated with the secondary antibody for 45 min on ice.

### Real-time polymerase chain reaction

Total RNA was extracted with the RNeasy Mini extraction kit (Qiagen, Germany) from cultured cells, cDNAs were prepared with an iScript cDNA Synthesis Kit (BioRad, Germany) and real-time polymerase chain reaction (PCR) was performed with an iCycler (BioRad). Each sample was measured in triplicate and values were normalized to *Gapdh*. Primer sequences for Lpxn and Pxn were from PrimerBank (*Spandidos et al., 2010*) (Lpxn: 26080416a1; aPxn: 114326500c2; bPxn: 22902122a1), GAPDH primers were described before (*Rognoni et al., 2014*) and Hic5 primers were newly designed (Hic5-fwd: 5'-ttcctttgcagcggttgttcc-3'; Hic5-rev: 5'-ggttacagaagccacatcgtggg-3').

### Antibodies and inhibitors

The following antibodies or molecular probes were used at indicated concentrations for western blot (WB), immunofluorescence (IF) or flow cytometry (FACS): kindlin-1 (home made), (*Ussar et al., 2008*) WB: 1:5000, IF: 1:1000; kindlin-2 (MAB2617 from Millipore, Germany) WB: 1:1000, IF: 1:500; talin (8D4 from Sigma, Germany) WB: 1:1000; talin (sc-7534 from Santa Cruz, Germany) IF: 1:500; talin-1 (ab57758 from Abcam, UK) WB: 1:2000; talin-2 (ab105458 from Abcam) WB: 1:2000; GAPDH (6C5 from Calbiochem, Billerica, MA, USA) WB: 1:10,000; Paxillin (610051 from BD Transduction Laboratories, Franklin Lakes, NJ, USA) WB: 1:1000, IF: 1:400; integrin β1-488 (102211 from Biolegend, San Diego, CA, USA) IF: 1:400, FACS: 1:200; integrin β1 (MAB1997 from Chemicon, Billerica, MA, USA) FACS: 1:400; integrin β1-647 (102213 from Biolegend) IF: 1:200; integrin β1 (home-made), (*Azimifar et al., 2012*) IF: 1:400; integrin β3-biotin (553345 from PharMingen, Franklin Lakes, NJ, USA) FACS: 1:200; integrin β3 (M031-0 from Emfret, Germany) IF: 1:200; integrin α2-FITC (554999 from BD Biosciences) FACS: 1:100; integrin α3 (AF2787 from R&D, Germany) FACS: 1:200; integrin α5-biotin (557446 from Pharmingen) FACS: 1:200, IP 1μg; integrin α5 (4705 from Cell Signaling, Germany) WB: 1:1000; integrin α6-FITC (555735 from Pharmingen) FACS 1:100; integrin αv-biotin (551380 from Pharmingen) FACS: 1:200; β1-integrin 9EG7 (550531 from BD Biosciences, San Diego, CA, USA) IF: 1:200; FACS: 1:200;

fibronectin (AB2033 from Millipore) IF: 1:500; IgG2a rat isotype control (13-4321 from eBioscience, Germany) FACS: 1:200; IP 1μg; Tritc-Phalloidin (P1951 from Sigma) IF: 1:400; Flag-tag-HRP (8592 from Sigma) WB: 1:10,000; GFP (A11122 from Invitrogen, Germany) WB: 1:2000; Cherry (PM005 from MBL, Woburn, MA, USA) WB:1:1000; Myc (05-724 from Millipore) WB 1:2000; FAK (06-543 from Upstate, Billerica, MA, USA) WB: 1:1000; FAK (3285 from Cell Signaling) WB (1:1000); phospho-Y397 FAK (3283 from Cell Signaling) WB: 1:1000; phospho-Y397 FAK (44624G from Biosource, Waltham, MA, USA) WB: 1:1000, IF: 1:400; ILK (611803 from Transduction Labs) WB: 1:5000; IF: 1:500; phospho-Y992 EGFR (2235 from Cell Signaling) WB: 1:2000; phospho-Y31 Paxillin (44720G from Invitrogen) WB: 1:1000; phospho-Y118 Paxillin (44722G from Invitrogen) WB: 1:1000; p130Cas (P27820 Transduction Labs) WB: 1:1000; phospho-Y410 p130 Cas (4011S from Cell Signaling) WB: 1:1000; Akt (9272 from Cell Signaling) WB: 1:1000; phospho-S473 Akt (4060 from Cell Signaling) WB: 1:1000; phosho-T308 Akt (9275 from Cell Signaling) WB: 1:1000; Erk1/2 (9102 from Cell Signaling) WB: 1:1000; Erk1/2 phosphorylated T202 Y204 (4376 Cell Signaling) WB: 1:1000.

The following secondary antibodies were used: goat anti-rabbit Alexa 488 (A11008), goat anti-mouse Alexa 488 (A11029), goat anti-rat Alexa 488 (A11006), goat anti-mouse Alexa 546 (A11003), donkey anti-mouse Alexa 647 (A31571), goat anti-rabbit Alexa 647 (A21244), (all from Invitrogen) FACS: 1:500, IF: 1:500; streptavidin-Cy5 (016170084) FACS: 1:400; goat anti-rat horseradish peroxidase (HRP) (712035150) (both from Dianova, Germany) WB: 1:10,000, donkey anti-rabbit Cy3 (711-165-152) (from Jackson ImmunoResearch, West Grove, PA, USA) IF: 1:500, goat anti-mouse HRP (172-1011) and goat anti-rabbit HRP (172-1019) (both from BioRad) WB: 1:10,000.

The FAK inhibitor PF-228 (PZ0117 from Sigma) was dissolved in dimethyl sulfoxide at 10 mM and used at 1:2000.

## Expression and purification of recombinant proteins

The recombinant expression of kindlin-2, full-length paxillin (paxillin-FL) and paxillin-LIM3 in *Escherichia coli* Rosetta cells (Merck Millipore) was induced with 1 mM or 0.2 mM IPTG, respectively, at 18°C for 22 hr. After cell lysis and clarification of the supernatant, kindlin-2 was purified by Ni-NTA affinity chromatography (Qiagen). Eluate fractions containing kindlin-2 were pooled, cleaved with SenP2 protease and purified by size-exclusion chromatography (Superdex 200 26/600, GE Healthcare, UK) yielding unmodified murine kindlin-2. The paxillin constructs were purified by Ni-NTA affinity chromatography (Qiagen), and subsequent size-exclusion chromatography (SEC650, BioRad) to obtain N-terminally tagged His10-SUMO3-paxillin-FL and His10-SUMO3-paxillin-LIM3 domain, respectively.

## Immunostaining

For immunostaining, cells were cultured on plastic ibidi-μ-slides (80826 from Ibidi, Germany) coated with 20 μg ml$^{-1}$ FN (Calbiochem). Cells were routinely fixed with 4% paraformaldehyde (PFA) (w/v) in phosphate buffered saline (PBS; 180 mM NaCl, 3.5 mM KCl, 10 mM Na$_2$HPO$_4$, 1.8 mM K$_2$H$_2$PO$_4$) for 10 min at room temperature (RT) or with –20°C cold acetone–methanol when indicated. If necessary, cells were solubilized with staining buffer (PBS supplemented with 0.1% Triton X-100 (v/v) and 3% BSA (w/v)) or with –20°C cold methanol for kindlin-2 staining. Background signals were blocked by incubating cells for 1 hr at RT in staining buffer. Subsequently, they were incubated in the dark with primary and secondary antibodies diluted in staining buffer. Fluorescent images were aquired with a LSM 780 confocal microscope (Zeiss, Germany) equiped with a 100×/NA 1.46 oil objective and with a DMIRE2-SP5 confocal microscope (Leica, Germany) equiped with a 40×/NA 1.25 or 63×/NA 1.4 oil objective using Leica Confocal software (version 2.5 build 1227). Brightfield images were aquired with an Axioskop (Carl Zeiss) 40×/NA 0.75 objective and DC500 camera with IM50 software (Leica). Z-stack projection and contrast adjustments ImageJ (v1.47) were used for further image analysis.

Super-resolution imaging was carried out by direct stochastic optical reconstruction microscopy (dSTORM) (*van de Linde et al., 2011*), which is based on precise emitter localization. To induce reversible switching of the Alexa 647 label and reduce photobleaching, imaging was performed in imaging solution (50% Vectashield (v/v) (H-1000; Vector Laboratories, Burlingame, CA, USA), 50% TBS (v/v), pH=8.0) supplemented with 50 mM β-mercaptoethylamine (Sigma-Aldrich; M9768).

dSTORM was implemented on a custom built total internal reflection fluourescence (TIRF) system (Visitron Systems, Germany) based on a Zeiss Axiovert 200M with fiber-coupled lasers. Sample were

excited with a 640 nm laser in a TIRF mode using a Zeiss α Plan-Fluar 100×/NA 1.45 oil objective. The emitted light was detected in the spectral range 660–710 nm through a Semrock FF02-685/40-25 bandpass filter (Semrock Inc., Rochester, NY, USA). Images were recorded with a Photometrics Evolve Delta emCCD camera (Photometrics, Huntington Beach, CA, USA), with its EM gain set to 250. Additional magnification by a factor of 1.6 resulted in a pixel size of 100 nm. For each final image, a total of 20,000 frames with an exposure time of 14 ms were recorded.

A standard TIRF imaging of the same sample in the green channel (anti-paxillin) was achieved by illumination with a 488 nm laser and detection in the spectral range 500–550 nm through a Chroma Et 525/50 bandpass filter (Chroma Technology Corporation, Bellows Falls, VT, USA). Simultaneous dual-colour imaging of both the green and the red channel was realized with a Hamamatsu W-View Gemini image splitter (Hamamatsu Photonics, Bridgewater, NJ, USA) mounted between the microscope and the camera. Image analysis was carried out with the ImageJ plugin ThunderSTORM (*Ovesny et al., 2014*) and standard tools of ImageJ. Heat maps of density of blink events were created using the 2D-Frequency Count/Binning module of OriginPro 9.1 (OriginLab Corporation, Northampton, MA, USA).

## AFM-based single-cell force spectroscopy

Tipless, 200 μm long V-shaped cantilevers (spring constants of 0.06 N m$^{-1}$; NP-O, Bruker, Billerica, MA, USA) were prepared for cell attachment as described (*Friedrichs et al., 2010*). Briefly, plasma cleaned cantilevers were incubated in 2 mg ml$^{-1}$ ConA (Sigma) in PBS at 4°C overnight. Polydime-thylsiloxan (PDMS) masks were overlaid on glass bottoms of Petri dishes (35 mm FluoroDish, World Precision Instruments, Sarasota, FL, USA) to allow different coatings of the glass surface (*Te Riet et al., 2014*). PDMS-framed glass surfaces were incubated overnight with 50 μg ml$^{-1}$ FN-RGD and 50 μg ml$^{-1}$ FN-ΔRGD in PBS at 4°C. Overnight serum-starved fibroblasts (Tln$^{Ctr}$, Kind$^{Ctr}$, Tln$^{Ko}$, Kind$^{Ko}$) grown on FN-coated (Calbiochem) 24 well plates (Thermo Scientific, Denmark) to confluency of ~ 80% were washed with PBS and detached with 0.25% (w/v) trypsin/EDTA (Sigma). Detached cells were suspended in single-cell force spectroscopy (SCFS) medium (DMEM supplemented with 20 mM HEPES) containing 1% (v/v) FCS, pelleted and further resuspended in serum-free SCFS medium. Detached cells were left suspended in SCFS media to recover from detachment for ~1 hr (*Schubert et al., 2014*). For the activation or chelation assay, the detached cells were incubated in SCFS media supplemented with 0.5 mM Mn$^{2+}$ or 5 mM EDTA, respectively, for ~1 hr and SCFS was performed in the presence of the indicated supplement. SCFS was performed using an AFM (Nano-Wizard II, JPK Instruments, Germany) equipped with a CellHesion module (JPK Instruments) mounted on an inverted optical microscope (Zeiss Axiovert 200M). Measurements were performed at 37°C, controlled by a PetriDish Heater (JPK Instruments). Cantilevers were calibrated using the equipartition theorem (*Hutter and Bechhoefer, 1993*).

To attach a single cell to the cantilever, cell suspensions were pipetted to the region containing the FN-ΔRGD coating. The ConA functionalized cantilever was lowered onto a single cell with a velocity of 10 μm s$^{-1}$ until reaching a contact force of 5 nN. After 5 s contact, the cantilever was retracted from the Petri dish by 50 μm and the cantilever-bound cell was left for incubation for >10 min. For adhesion experiments, the cantilever-bound cell was brought into contact with the FN-Δ RGD coated support at a contact force of ~2 nN for 5, 20, 50 and 120 s and then retracted while measuring the cantilever deflection and the distance travelled. Subsequently, the cell adhesion to the FN-RGD coated support was characterized as described. In case cantilever attached cells showed morphological changes (e.g. spreading) they were discarded. The approach and retract velocity of the cantilever was 5 μm s$^{-1}$. The deflection of the cantilever was recorded as force-distance curves. Adhesion forces were extracted from retraction force-distance curves using the AFM data processing software (JPK Instruments).

## Immunoprecipitations and recombinant protein pulldown

GFP-IPs were performed using μ-MACS anti-GFP magnetic beads (130-091-288 from Miltenyi, Germany). To pulldown recombinant kindlin-2 35 μg of purified His10-LIM3 or 10 μg of purified His10-paxillin-FL were incubated with 100 μl of 50% Ni-NTA-Agarose slurry (Qiagen) in pulldown buffer (20 mM Tris, pH 7.5, 200 mM NaCl, 1 mM TCEP, 0.05% Tween20) for 1 hr at 4°C. After a first wash with 20 column volumes (CV) of pulldown buffer supplemented with 1 mM ZnCl$_2$ and a second wash with

20 CV of pulldown buffer, 14 µg of purified kindlin-2 were added to 100 µl of Ni-NTA-agarose slurry and incubated for 30 min at 4°C. Subsequently, the Ni-NTA beads were washed three times with 20 CV of pulldown buffer supplemented with 25 mM imidazole and either 1 mM $ZnCl_2$ or 1 mM EDTA. The beads were eluted with 50 µl pulldown buffer supplemented with 500 mM imidazole and analysed on a 12% sodium dodecyl sulfate polyacrylamide gel electrophoresis (SDS-PAGE).

For immunoprecipation of kindlin-2 or paxillin, control fibroblasts were lysed in lysis buffer (50 mM Tris, pH 8.0, 150 mM NaCl, 1% Triton X-100, 0.05% sodium deoxycholate, 10 mM EDTA). Lysates were incubated with kindlin-2 or paxillin antibodies for 2 hr at 4°C while rotating. Isotype-matched IgG was used as a negative control. After this, lysates were incubated with 50 µl protein A/G Plus Agarose (Santa Cruz) for 2 hr at 4°C. Following repeated washes with lysis buffer, proteins were eluted from the beads using Laemmli buffer and analyzed by western blotting.

For the immunoprecipitation of α5 integrin from the cell surface of live cells, α5 integrins were labeled with a biotinylated anti-α5 integrin antibody (PharMingen #557446) or an isotype control (eBioscience # 13-4321) for 1 hr on ice. After two washes in ice-cold PBS to remove unbound antibody, cells were lysed in IP buffer (50 mM Tris, pH 7.5, 150 mM NaCl, 1% Triton X-100, 0.1% sodium deoxycholate, 1mM EDTA, and protease inhibitors) and cleared by centrifugation. α5 integrin immuno-complexes were pulled-down by incubation with streptavidin-sepharose (GE Healthcare) overnight at 4°C with gentle agitation. After several washes with lysis buffer, proteins were subjected to SDS-PAGE and western blot analysis with antibodies against α5 and β1 integrin.

## Spreading and adhesion assays

Cells were grown to 70% confluency and then detached using trypsin/EDTA. Suspended cells were serum starved for 1 hr in adhesion assay medium (10 mM HEPES, pH 7.4; 137 mM NaCl; 1 mM $MgCl_2$; 1 mM $CaCl_2$; 2.7 mM KCl; 4.5 g $L^{-1}$ glucose; 3% BSA (w/v)) before 40,000 cells per well were plated out in the same medium supplemented with 8% FCS, and 5 mM $Mn^{2+}$ if indicated. Plastic ibidi-µ-slides (Ibidi; 80826) were coated with 10 µg $ml^{-1}$ FN (Calbiochem) for adhesion or 20 µg $ml^{-1}$ FN for spreading assays, 10 µg $ml^{-1}$ LN (11243217001 from Roche, Germany), 10 µg $ml^{-1}$ COL (5005B from Advanced Bio Matrix, Carlsbad, CA, USA), 10 µg $ml^{-1}$ VN (07180 from StemCell, Canada) or 0.1% PLL (w/v) (Sigma; P4707) diluted in PBS. Seeded cells were centrifuged at 600 rpm in a Beckman centrifuge for 30 min at 37°C before they were fixed with 4% PFA (w/v) in PBS and stained with Phalloidin-TRITC and DAPI. For cell adhesion assays, nuclear staining of the whole well was imaged using a 2.5x objective and cell numbers were counted using ITCN plugin for imageJ (*Byun et al., 2006*). For cell spreading assays, 12 confocal images of different regions of Phalloidin and DAPI stained cells were aquired using a Leica confocal microscope, cell spreading was quantified using imageJ.

For time dependent and ligand concentration dependent adhesion on FN, 40,000 cells were plated on 96-well plates, vigorously washed after the indicated timepoints with PBS and fixed with 4% PFA. Cell attachment was meassured by crystal violet staining (0.1% in 20% methanol) of cells in a absorbance plate reader at the wavelength of 570 nm.

## Live cell imaging

A hole of 15 mm diameter was drilled into the bottom of a 35 mm falcon tissue culture dish (353001, Becton Dickinson) and a coverslip (Ø 25 mm, Menzel-Gläser, Germany), rinsed with ethanol, was glued to the dish with silicon glue (Elastosil E43, Wacker, Germany). After coating coverslips with 20 µg $ml^{-1}$ FN (Calbiochem) overnight at 4°C, cells were plated and imaged in an inverted transmission light microscope (Zeiss Axiovert 200 M, Carl Zeiss) equipped with a climate chamber. Phase contrast images were taken with a ProEM 1024 EMCCD camera (Princeton Instruments, Acton, MA, USA) through a Zeiss Plan Neofluar 100x objective (NA 1.3, Ph3). Frames were acquired at 30 sec or 1 min intervals and converted to time lapse movies using ImageJ.

## Constructs and transfections

K2ΔPHGFP was cloned by PCR using the K2GFP cDNA (*Ussar et al., 2006*) as template and the Kind2fwd (5'-ctcgaggaggtatggctctggacgggataag -3', Kind2PHrev 5'-**tggtcttgcctttaatatag**tcagcaagtt -3'), Kind2PHfwd (5'-**ctatattaaaggcaa**gaccatggcagacag -3') and Kind2rev (5'- tctagatcacacccaac-cactggtga-3') primers. The two fragments containing homologous regions (indicated with bold

letters in the primer sequences) were fused by another round of amplification using the most 5' and 3' primers (Kind2fwd and Kind2rev). The resulting PCR product was cloned into the K2GFP vector. The N- and C-terminal truncation constructs of kindlin-2 were cloned by PCR using K2GFP as template. The primer sequences were: Kind2-NT-fwd 5'-ctgtacaagtccggactc-3', Kind2-NT-rev 5'-gcggccgcctattttgctttatcaagaagagc-3', Kind2-CT-fwd 5'-ctcgagctatggataaagcaaaaaccaaccaag-3', Kind2-CT-rev 5'-gttatctagagcggccgc-3'. Stable expression of K2ΔPHGFP and FAKGFP- or Myc-FAK (a gift from Dr. Ambra Pozzi; Vanderbilt University, Nashville, USA) cDNAs was achieved with the sleeping beauty transposase system (*Bottcher et al., 2012*). Kindlin-1-GFP and Kindlin-3-GFP constructs have been described (*Ussar et al., 2008*; *Moser et al., 2008*).

For stable expression of murine talin-1 and THD (amino acids1-443), the corresponding cDNAs were N-terminally tagged with Venus and cloned into the retroviral pLPCX vector. The constructs for GFP-tagged paxillin-LIM truncation mutants were generated by PCR from GFP- and Cherry-tagged α-paxillin (*Moik et al., 2013*) and cloned into the retroviral pLPCX vector. The primer sequences were: stop codon in bold: ΔLIM1-4fwd 5'- caccgttgccaaa**tga**gggtctgtggagcc -'3, ΔLIM1-4rev 5'-ggctccacagaccc**tca**tttggcaacggtg -'3, ΔLIM2-4fwd 5'- cagcctcttctcccca**tga**cgctgctactactg -'3, ΔLIM2-4rev 5'- cagtagtagcagcg**tca**tggggagaagaggctg -'3, ΔLIM3-4fwd 5'- aagattacttcgacatgtttgct**tga**cccaagtgcggc -'3, ΔLIM3-4rev 5'- gccgcacttggg**tca**agcaaacatgtcgaagtaatctt -'3, ΔLIM4fwd 5'-ggcgcggctcg**tga**ctgtgctccgg -'3, ΔLIM4rev 5'- ccggagcacag**tca**cgagccgcgcc -'3). The cDNA of murine Hic5 was amplified from a cDNA derived from murine vascular smooth muscle cells, cloned into pCR2.1-TOPO (Invitrogen) and subcloned into pEGFP-C1 vector. Murine leupaxin cDNA (cloneID: 5065405 from Thermo Scientific, Germany) was PCR-amplified (Lpxn-fwd: 5'- ctcgagcaatggaagagctggatgccttattg -3'; Lpxn-rev 5'- gaattcctactgtgaaaagagcttagtgaagc -3') and subcloned into the pEGFP-C1 vector.

To express recombinant murine kindlin-2 and paxillin-LIM3 (A473-S533) cDNAs were fused with N-terminal tandem tags consisting of 10x-Histidine followed by a SUMO3-tag and cloned into pCoofy17. The primer sequences for amplifying the paxillin-LIM3 domain were: LIM3fwd 5'-aaccggtggagctcccaagtgc-3' and LIM3rev 5'-ttctcgagttacgagccgcgcc-3'. The plasmid carrying $FNIII_{7-10}$ cDNA has been described previously (*Takahashi et al., 2007*). For Y2H analysis, the kindlin-2 cDNA was PCR amplified using the primers K2-Bamfw: 5'-gggatcccactgggcctaatggctctggacgggataagg-3' and K2-Salrev: 5'-gtgtcgacgtcacacccaaccactggtgagtttg-3' and cloned into the pGBKT7 plasmid to obtain a kindlin-2 version that was N-terminally fused with the Gal4-DNA binding domain. Screening of this construct against a human full ORF library was conducted by the Y2H protein interaction screening service of the German Cancer Research Center in Heidelberg, Germany.

## Statistical analysis

Experiments were routinely repeated at least three times and the repeat number was increased according to the effect size or sample variation. Unless stated differently, all statistical significances (*$P<0.05$; **$P<0.01$; ***$P<0.001$; n.s., not significant) were determined by two-tailed unpaired $t$-test. In the boxplots, the middle line represents the median, the box ends represent the 25th and 75th percentiles and the whisker ends show the 5th and 95th percentiles. Statistical analysis were performed with Prism (GraphPad, La Jolla, CA, USA).

## Acknowledgements

We thank Ursula Kuhn for expert technical help. The work was supported by RO1-DK083187, RO1-DK075594, R01-DK069221 and VA Merit Award 1I01BX002196 (to RZ), by the Deutsches Zentrum für Herz-Kreislauf-Forschung, partner site Munich Heart Alliance (to RTB and RF) and by the European Research Council (Grant Agreement no. 322652), Deutsche Forschungsgemeinschaft (SFB-863) and the Max Planck Society (to RF).

# Additional information

## Funding

| Funder | Grant reference number | Author |
| --- | --- | --- |
| Veterans Affair Merit Award | 1I01BX002196 | Roy Zent |
| NIH Office of the Director | RO1-DK083187 | Roy Zent |
| NIH Office of the Director | RO1-DK075594 | Roy Zent |
| NIH Office of the Director | R01-DK069221 | Roy Zent |
| European Research Council | 322652 | Reinhard Fässler |
| Deutsche Forschungsgemeinschaft | SFB-863 | Reinhard Fässler |
| Max-Planck-Gesellschaft | | Reinhard Fässler |

The funders had no role in study design, data collection and interpretation, or the decision to submit the work for publication.

## Author contributions

MT, MW, Carried out most experiments and data analysis, evaluated and interpreted the data.; RTB, Tested integrin activation, paxillin recruitment and the kindlin-2 interaction with paxillin in vivo, evaluated and interpreted the data.; ER, Performed immunoblottings and PCR experiments; MV, Produced recombinant kindlin-2 and paxillin.; MB, Performed AFM experiments; AL, Performed dSTORM; KA, Analyzed FN assembly; DJM, Evaluated and interpreted the data; RZ, Crossed talin mice, established cell lines, evaluated and interpreted the data.; RF, Initiated, conceived and directed the project, evaluated and interpreted the data and wrote the manuscript.

## Ethics

Animal experimentation: Housing and use of laboratory animals at the Max Planck Institute of Biochemistry are fully compliant with all German (e.g. German Animal Welfare Act) and EU (e.g. Annex III of Directive 2010/63/EU on the protection of animals used for scientific purposes) applicable laws and regulations concerning care and use of laboratory animals. All of the animals were handled according to approved license (No.5.1-568- rural districts office). The animal experiments using the talin mice were performed with the approval of the Vanderbilt Institute Animal Care and Use Committee under protocol M09/374.

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
