## [Decision Letter]

Thank you for submitting your work entitled "Kindlin-2 cooperates with talin to activate integrins and induces cell spreading by directly binding paxillin" for peer review at *eLife*. Your submission has been favorably evaluated by Vivek Malhotra (Senior editor) and three reviewers.

The reviewers have discussed the reviews with one another and the Senior editor has drafted this decision to help you prepare a revised submission.

Summary:

The authors provide evidence in support of the hypothesis that talins and kindlins have a role in the activation of fibronectin-binding integrins in mesenchymal cells. They also identify a potential novel mechanism through which fibronectin-binding integrins can activate FAK and promote spreading and proliferative and survival signalling via formation of lamellipodia.

These findings are highly significant, however, the authors need to show that kindlin regulates integrin's affinity for its ligands and identify the signalling pathways regulated by kindlin and their functional significance. The specific major (essential) revisions and minor issues of the reviewers follow.

Essential revisions:

1) The results in Figure 2 are really interesting but, rather than measuring adhesion to fibronectin, the authors should measure the binding of the soluble cell binding domain of fibronectin and of Mab 9EG7 to suspended cells +/- Mn^2+^. Expression of Rap1-V12 from an inducible promoter can be used to "physiologically" activate integrins.

2) The proposed proliferative defect shown in Figure 1—figure supplement 3 is just a modest delay in entry in S phase. If the cells were just plated, it may be a consequence of the delayed spreading. Growth curves over 1 week would be necessary to strengthen the contention that kindlins promote cell proliferation. In addition, the proposed survival defect shown in Figure 1—figure supplement 3 is non-existent as the talin and the kindlin KO cells survive very well (only 0.5 to 1% of them undergo apoptosis). Importantly, the immunoblotting experiments in Figure 5 are superficial. They need to be extended to include cells not treated with Mn^2+^ as well as talin KO cells reconstituted with talin. In addition, the phosphorylation of paxillin, Akt at residue T408 (the required PDK1 site), p70-S6K, S6, 4E-BP1, Raf, MEK and ERK has not been studied. Finally, that FAK kinase activity necessarily plays a role in signaling downstream of kindlins is not convincing as PF-228 is used at 5 μM (25x dose required to inhibit FAK 397 in most cells).

3) The contention that loss of talins or kindlins changes the repertoire of fibronectin-binding integrins in a similar fashion (Figure 2) is not convincing. The data seem to suggest that there is a switch from α-v β-3 to α-v β-5 and that the decreased expression of α-3 and α-6 allows equal expression of α-5 in kindlin-null cells, but the changes may be more complex as the authors have not examined the expression of α-v β-1. The most definitive way to address the issue is to purify fibronectin-binding integrins from surface-labeled cells by affinity chromatography on Fn-CBD in the presence of Mn^2+^ and subject the eluted proteins to IP with anti-α-v, α-5, β-1, β-3, and β-5.

4) The sections describing effects of talin and kindlin on integrin "activation" are quite confusing, although it is clear that talin and kindlin are important for adhesion to Fn. Mn^2+^ activation clearly does partially rescue the adhesion defect of both talin and kindlin deficient cells, but the AFM profiles for talin and kindlin deficient cells look completely different in the presence of Mn^2+^. This should be emphasized in the text. Is it assumed that only the first 5s AFM time point assesses "activation" and later timepoints assesses "integrin clustering"? It is difficult to see from the AFM figures that Mn^2+^ induced "stronger integrin-ligand bonds" than non-treated cells (Figure 2), as the controls (non-Mn^2+^ treated cells) are shown in Figure 2. If the authors assume the same degree of integrin activation in talin and kindlin KO cells in the presence of Mn^2+^, this should be examined directly. To assess activation, effects on 9EG7-binding in talin and kindlin deficient cells vs controls in the absence and presence of Mn^2+^, should be quantified, for example by flow cytometry. In addition, soluble binding assays should be performed (as the authors have done previously in Montanez et al, 2008). The authors have attempted to assess clustering of the integrins in talin KO and kindlin KO cells in Figure 3. It is clearly difficult to assess specific vs spatially-driven clustering in cells with such different morphology, and therefore difficult to draw conclusions about the role of kindlin in integrin clustering in these cells. In the Results section, the authors state that "the integrin-activating compound Mn^2+^ can only partially substitute for the adhesion promoting roles that talin and kindlin accomplish together", however, it is stated that "we circumvented their integrin activation defect with Mn^2+^". If there is a severe activation/adhesion defect in the presence of Mn^2+^ (especially for Kindlin-KO cells), this clearly also has implications for outside-in signaling profiles the cells.

5) It is unclear from the results that the integrin/paxillin complex promotes cell proliferation, as stated in the subheading “The kindlin-2/paxillin complex promotes FAK-mediated cell spreading and proliferation”; this has not directly been assessed here. Also, the statement about a connection between the reduced Akt signaling and proliferation "which provides an explanation for their reduced proliferation…" is too strong, as this hypothesis has not been tested here. Kindlin deletion has also been shown to affect other signaling pathways such as TGF-β signaling and proliferation, etc. Do cells which express kindlin that cannot bind paxillin display reduced Akt activation and proliferation? Does expression of active FAK influence Akt activation or proliferation? Does the cell survival defect of kindlin and talin KO cells unspecifically affect their adhesion and signaling profiles?

6) As kindlin clearly has other effects in cells than those dependent on integrins, it would be important to assess whether cells where the integrin-kindlin interaction has been abolished show the same phenotype as KO cells. E.g., what is the adhesion and signaling phenotype of TT/AA-β1-integrin fibroblasts (where kindlin cannot bind the integrin), do they have similar defects in spreading, FAK activation, signaling and proliferation as kindlin knockout cells? What about cells expressing mutant kindlin that cannot bind integrin?

7) Integrin-mediated adhesion depends on the level of integrin expression. Since integrins are heterodimers, we didn't find measurements of individual integrin subunits reassuring that levels of the relevant integrin for these studies, α5β1 (or αvβ3 for that matter), were comparable among the various cells. Flow cytometry using complex-specific antibodies would address this concern.

8) Since the KO cells express αvβ3, as well as α5β1, the inability of Mn^2+^ have any effect on cell adhesion to vitronectin is puzzling.

9) Considering the variability in the AFM measurements, it appears that in Figure 2, in contrast to Kind^Ko^ cells, the Tln^Ko^ cells do develop measureable adhesions to FN and that Mn^2+^ increases the strength of the adhesions of nearly control levels. Similarly, Mn^2+^ increased the adhesion of the Kind^Ko^ cells, although less so than the control cells. Since there is evidence that the interaction of integrins with ligands is a multi-step process with adhesion strength increasing as the time of contact between integrin and ligand increases, the interpretation of the these experiments, i.e., the kindlin stabilizes integrin ligand complexes with time by inducing or maintaining integrin clusters is not necessarily the only conclusion.

10) Figure 3: If we have interpreted the distribution of dots correctly, these data suggest that talin and kindlin are not necessarily required to form integrin clusters.

11) If the control fibroblasts are maintained in suspension, are kindlin and talin constitutively associated with β1 tail? Do the cells constitutively bind 93G7? What is the distribution of integrin in the suspended cells?

[Editors' note: further revisions were requested prior to acceptance, as described below.]

Thank you for resubmitting your work entitled "Kindlin-2 cooperates with talin to activate integrins and induces cell spreading by directly binding paxillin" for further consideration at *eLife*. Your revised article has been favorably evaluated by Vivek Malhotra (Senior editor) and three reviewers. The manuscript has been improved but there are some remaining issues that need to be addressed before acceptance, as outlined below:

Specifically, the loading controls for Erk, Akt and Cas in kindlin control and KO cells show very variable levels of expression of these proteins in the different cells. Please provide a new gel to address this minor issue.

---

## [Author Response]

Essential revisions: 1) The results in Figure 2 are really interesting but, rather than measuring adhesion to fibronectin, the authors should measure the binding of the soluble cell binding domain of fibronectin and of Mab 9EG7 to suspended cells +/- Mn^2+^. Expression of Rap1-V12 from an inducible promoter can be used to "physiologically" activate integrins.

We measured the binding of the soluble cell-binding domain of fibronectin (FN) and of Mab 9EG7 to suspended cells +/- Mn^2+^ by FACS.

The new data analyzing 9EG7 binding to the Tln^Ctr^, Kind^Ctr^, Tln^Ko^ and Kind^Ko^ cells and the rescue cells is included as new Figure 2—figure supplement 3. These data confirm that talin and kindlin are required to activate β1 integrin and to stabilize Mn^2+^-induced unbending/unclasping of α5β1 integrins.

We also measured binding of labeled soluble cell binding domain of FN and observed weak binding of soluble FN to suspension cells that is increased upon Mn^2+^ treatment. Although we can quantify these differences, the shifts in the FACS histograms are very small and almost overlapping with background binding (Figure 7). We believe that this renders the analysis unreliable as small shifts lead to a dramatic relative increase, which may not reflect the reality. For this reason we decided against including this data into the revised manuscript.

Author response image 1.FN binding by Tln^Ko^ and Kind^Ko^ cells.Left: Quantification of binding of soluble, fluorescently labelled FN-RGD relative to the FN-RGE peptides; the FN-RGD/FN-RGE ratio of pKO cells under control conditions was set to 1 and subtracted from all other results. Binding was measured in the presence of 1 mM MgCl_2_ (Ctr) or 5 mM MnCl_2_ (Mn^2+^) (independent experiments: n=10 for Kind and Tln cell lines, n=3 for pKO; error bars indicate SEM; significances from ratio paired two tailed t-test are given for pairs connected with brackets). Right: Individual FACS histograms of the indicated cell lines used for quantification of fluorescently labelled FN-RGD binding.**DOI:**
http://dx.doi.org/10.7554/eLife.10130.029

We did not test the ability of Rap1-V12 to “physiologically” activate integrins due to time constraints and because Rap1-V12 was mainly used to activate integrins on hematopoietic cells.

2) The proposed proliferative defect shown in Figure 1—figure supplement 3 is just a modest delay in entry in S phase. If the cells were just plated, it may be a consequence of the delayed spreading. Growth curves over 1 week would be necessary to strengthen the contention that kindlins promote cell proliferation. In addition, the proposed survival defect shown in Figure 1—figure supplement 3 is non-existent as the talin and the kindlin KO cells survive very well (only 0.5 to 1% of them undergo apoptosis). Importantly, the immunoblotting experiments in Figure 5 are superficial. They need to be extended to include cells not treated with Mn^2+^ as well as talin KO cells reconstituted with talin. In addition, the phosphorylation of paxillin, Akt at residue T408 (the required PDK1 site), p70-S6K, S6, 4E-BP1, Raf, MEK and ERK has not been studied. Finally, that FAK kinase activity necessarily plays a role in signaling downstream of kindlins is not convincing as PF-228 is used at 5 μM (25x dose required to inhibit FAK 397 in most cells).

The two aims of this paper were (1) to investigate whether talin and kindlin are as essential for integrin-mediated adhesion of fibroblasts as for integrin-mediated adhesion of hematopoietic cells (neutrophils, T cells, platelets, hematopoietic stem and progenitor cells, etc.), and (2) to determine what kind of major signaling outputs can be delivered by talin or kindlin. We found that (1) talin and kindlin are essential to induce integrin-mediated adhesion of fibroblasts to FN, and that (2) kindlin can induce isotropic spreading by recruiting paxillin. The beauty of the latter finding was that it solved a conundrum and showed for the first time a mechanism explaining how paxillin (which was discovered in the early nineties) can be recruited to integrin adhesion sites.

We also noted a slight proliferation defect in cells lacking kindlin expression. Although this is in agreement with a reduced FAK activation triggered by paxillin-mediated recruitment of FAK to integrin adhesion sites, we never aimed at deeply investigating this finding, mainly because our cells were immortalized and therefore not suitable for such studies. The same is true for the survival. We therefore removed the cell survival data from the manuscript and toned down our findings on cell proliferation in the text.

In parallel we performed a number of experiments suggested by the reviewers:

a) We replaced the BrdU proliferation assay with growth curves over 5 days that show the reduced proliferation of Kind^Ko^ cells compared to Talin^Ko^ and the control cells (Figure 1—figure supplement 3).

b) We repeated the immunoblots in Figure 5 including both rescue cell lines and different conditions (FN, FN+EGF, FN+EGF+Mn^2+^) and probed with antibodies against different signalling components including Y410-p130Cas, pT308-Akt, S473-Akt, pT202/pY204 Erk1/2, pY31-Pxn, pY118-Pxn (Figure 5 and Figure 5—figure supplement 3).

c) Regarding the FAK kinase activity, we repeated the assay using increasing concentrations of FAK inhibitor starting from 0.3 µM, which show a dose-dependent decrease in the percentage of lamellipodia in Tln^Ko^ cells (Figure 5).

3) The contention that loss of talins or kindlins changes the repertoire of fibronectin-binding integrins in a similar fashion (Figure 2) is not convincing. The data seem to suggest that there is a switch from α-v β-3 to α-v β-5 and that the decreased expression of α-3 and α-6 allows equal expression of α-5 in kindlin-null cells, but the changes may be more complex as the authors have not examined the expression of α-v β-1. The most definitive way to address the issue is to purify fibronectin-binding integrins from surface-labeled cells by affinity chromatography on Fn-CBD in the presence of Mn^2+^ and subject the eluted proteins to IP with anti-α-v, α-5, β-1, β-3, and β-5.

The reviewers are correct; we indeed observe a switch from αvβ3 to αvβ5 in the kindlin and talin knockout cell lines. Our whole studies rely on similar expression of α5 integrins. Our initial FACS analyses show this and we now also determined the levels of α5β1 integrin by selective immunoprecipitation of α5 integrins from the cell surface of live cells of the talin and kindlin control and knockout cell lines followed by western blot analysis with antibodies against α5 and β1 integrins. These experiments show comparable levels of α5β1 integrin presented on the cell surface of Talin^Ctr^, Kind^Ctr^, Talin^Ko^ and Kind^Ko^ cell lines. This data is included in the revised manuscript as Figure 2—figure supplement 2. In addition, we examined the expression of αvβ1 in our cell lines by αv immunoprecipitation followed by β1 western blotting and found either barely or no detectable β1 subunits. Altogether this indicates that the major FN-binding integrins on our cell lines are indeed α5β1 and αvβ5 integrins.

4) The sections describing effects of talin and kindlin on integrin "activation" are quite confusing, although it is clear that talin and kindlin are important for adhesion to Fn. Mn^2+^ activation clearly does partially rescue the adhesion defect of both talin and kindlin deficient cells, but the AFM profiles for talin and kindlin deficient cells look completely different in the presence of Mn^2+^. This should be emphasized in the text. Is it assumed that only the first 5s AFM time point assesses "activation" and later timepoints assesses "integrin clustering"? It is difficult to see from the AFM figures that Mn^2+^ induced "stronger integrin-ligand bonds" than non-treated cells (Figure 2), as the controls (non-Mn^2+^ treated cells) are shown in Figure 2. If the authors assume the same degree of integrin activation in talin and kindlin KO cells in the presence of Mn^2+^, this should be examined directly. To assess activation, effects on 9EG7-binding in talin and kindlin deficient cells vs controls in the absence and presence of Mn^2+^, should be quantified, for example by flow cytometry. In addition, soluble binding assays should be performed (as the authors have done previously in Montanez et al, 2008). The authors have attempted to assess clustering of the integrins in talin KO and kindlin KO cells in Figure 3. It is clearly difficult to assess specific vs spatially-driven clustering in cells with such different morphology, and therefore difficult to draw conclusions about the role of kindlin in integrin clustering in these cells. In the Results section, the authors state that "the integrin-activating compound Mn^2+^ can only partially substitute for the adhesion promoting roles that talin and kindlin accomplish together", however, it is stated that "we circumvented their integrin activation defect with Mn^2+^". If there is a severe activation/adhesion defect in the presence of Mn^2+^ (especially for Kindlin-KO cells), this clearly also has implications for outside-in signaling profiles the cells.

The reviewers’ comments that the sections describing the effects of talin and kindlin on integrin ‘activation’ are confusing, although it is clear that talin and kindlin are important to establish cell adhesion to fibronectin. We apologize for the confusion and revised our text to describe these effects more clearly. The reviewers also mentioned that the AFM-based SCFS results show different adhesion profiles for talin and kindlin deficient cells in the presence of Mn^2+^. We agree that this is the case and now in our revision specifically report that the adhesion strength restored upon Mn^2+^ administration differs in both Tln^Ko^ and Kind^Ko^ cells. In our opinion, this result suggests that both talin and kindlin play different roles in integrin mediated adhesion wherein only the role of talin can be accomplished partially upon Mn^2+^ administration and not the role of kindlin.

The reviewers further question whether “it is assumed that only the first 5s AFM time point assesses "activation" and later time points assesses "integrin clustering". Yes, the first 5s AFM time point assesses very early adhesion events (i.e. "activation") established by integrins binding to FN. At later time points (>5s) cell adhesion is considerably strengthens and therefore may reflect the clustering of integrins. Such time dependency of cells establishing adhesion to ECM proteins has been frequently reported in SCFS experiments. The early binding events at short contact times between cell and substrate (within few seconds) are dominated by the binding of single integrins (or CAMs). At extended time ranges (above a few tens of seconds) the integrins (or CAMs) cluster thereby strengthening cell adhesion to the substrate (Taubenberger et al., Mol Biol Cell. 2007; Friedrichs et al., JBC 2008). Thus from insight provided previously we can safely assume that the cell adhesion increasing with higher contact times is due to integrin clustering.

The reviewers also comment that “it is difficult to see from the AFM figures that Mn^2+^ induced "stronger integrin-ligand bonds" than non-treated cells (Figure 2), as the controls (non-Mn^2+^ treated cells) are shown in Figure 2”. We agree, but used this representation, as we wanted to lay emphasis on differences in adhesion strengths among these cells with and without Mn^2+^.

As suggested by the reviewers we assessed integrin activation by analyzing the binding of Mab 9EG7 to wild-type and mutant fibroblasts in the presence and absence of Mn^2+^ by FACS. This data shows that talin as well as kindlin are required for activating β1 integrin and stabilizing Mn^2+^-induced unbending/unclasping of α5β1 integrins (Figure 2—figure supplement 3). We also determined the binding of soluble FN. Although we observed weak binding of soluble FN to suspension cells that is increased upon Mn^2+^ treatment we decided against including it into the revised manuscript because a detailed analysis of the FACS histograms revealed only minor shifts compared to background binding that renders the analysis unreliable (see also comment 1)

(c) We rephrased the sentence “we circumvented their integrin activation defect with Mn^2+^”. We wanted to say that the complete loss of integrin functions in the absence of either talin or kindlin required a trick that bypasses their role in integrin activation. Although Mn^2+^ is doing this, the effect of Mn^2+^ on mutant cells is significantly less pronounced than compared to wild-type cells indicating that the maintenance of Mn^2+^-induced integrin activation is also dependent on talin and kindlin. Hence, the two proteins are required to activate integrins (which we now show with binding of Mab 9EG7) and to maintain the “Mn^2+^-induced activation state”.

5) It is unclear from the results that the integrin/paxillin complex promotes cell proliferation, as stated in the subheading “The kindlin-2/paxillin complex promotes FAK-mediated cell spreading and proliferation”; this has not directly been assessed here. Also, the statement about a connection between the reduced Akt signaling and proliferation "which provides an explanation for their reduced proliferation…" is too strong, as this hypothesis has not been tested here. Kindlin deletion has also been shown to affect other signaling pathways such as TGF-β signaling and proliferation, etc. Do cells which express kindlin that cannot bind paxillin display reduced Akt activation and proliferation? Does expression of active FAK influence Akt activation or proliferation? Does the cell survival defect of kindlin and talin KO cells unspecifically affect their adhesion and signaling profiles?

As stated above (see point 2) we were not aiming at analyzing the role of kindlin and talin for proliferation and survival in depth. Our studies show an involvement of the two proteins but the changes between them are small, probably because our cells are immortalized with the large T antigen. We thank the reviewers for their comment and as a result we removed the part on survival and toned down the results of proliferation and stuck to our major aims: (1) do integrins work without talin or kindlin in non-hematopoietic cells? and (2) which major signaling outputs are triggered by FN-adherent cells that express either talin or kindlin (paxillin recruitment leading to membrane protrusion)?

As suggested by the reviewers, we still analyzed Akt activation in cells expressing Kindlin-2ΔPH, which shows impaired paxillin binding. Indeed we found that Kind^Ko^ cells re-expressing Kindlin2-GFP but not Kindlin-2ΔPH induce FAK and Akt phosphorylation when seeded on FN (Figure 5).

6) As kindlin clearly has other effects in cells than those dependent on integrins, it would be important to assess whether cells where the integrin-kindlin interaction has been abolished show the same phenotype as KO cells. E.g., what is the adhesion and signaling phenotype of TT/AA-β1-integrin fibroblasts (where kindlin cannot bind the integrin), do they have similar defects in spreading, FAK activation, signaling and proliferation as kindlin knockout cells? What about cells expressing mutant kindlin that cannot bind integrin?

The TT/AA mutation in the β1 integrin tail reduces but does not abolish kindlin-2 binding (Fitzpatrick et al., JBC 2014). Furthermore, cells expressing TT/AAβ1 integrins express several other integrins, which are not mutated and therefore bind kindlin. Similarly, mutant forms of kindlins reduce the binding affinities for integrins but do not abolish binding. This, however, would be important to evaluate the experiments with such mutants especially in light of a recent paper from our lab showing that minimal amounts of kindlin-3 are sufficient for basal cellular functions (Klapproth et al., Blood 2015). So, it is difficult to perform conclusive experiments with such cells (which are available in the lab) as this cells system is not ‘clean’ enough.

7) Integrin-mediated adhesion depends on the level of integrin expression. Since integrins are heterodimers, we didn't find measurements of individual integrin subunits reassuring that levels of the relevant integrin for these studies, α5β1 (or αvβ3 for that matter), were comparable among the various cells. Flow cytometry using complex-specific antibodies would address this concern.

We are not aware of complex-specific antibodies for mouse α5β1 integrins. In order to address the reviewers’ concern we determined the levels of α5β1 integrin by selective immunoprecipitation of α5 integrins from the cell surface of live Tln^Ctr^, Kind^Ctr^, Tln^Ko^ and Kind^Ko^ cells followed by western blot analysis with antibodies against α5 and β1 integrins. These experiments confirm the comparable levels of α5β1 integrin presented on the cell surface of Talin^Ctr^, Kindlin^Ctr^, Talin^Ko^ and Kindlin^Ko^ cell lines. This data is included in the revised manuscript as Figure 2—figure supplement 2. If at all, the Kindlin^Ko^ cells have even slightly more α5β1 integrin on their cell surface compared to Talin^Ko^ cells indicating that the cell adhesion and spreading defects of Kindlin^Ko^ cells on fibronectin are clearly not due to reduced α5β1 integrin levels.

8) Since the KO cells express αvβ3, as well as α5β1, the inability of Mn^2+^ have any effect on cell adhesion to vitronectin is puzzling.

We thank the reviewers for this comment. We carefully analyzed our vitronectin and realized by SDS-PAGE that our vitronectin batch was faulty as we observed protein running at the wrong molecular weight. We repeated the adhesion assay with different vitronectin batches form different companies and can now show adhesion of Mn^2+^-treated Tln^Ko^ but not Kind^Ko^ cells to VN. The new adhesion assay is shown in Figure 1. The differential adhesion of Mn^2+^-treated Tln^Ko^ and Kind^Ko^ cells to VN (Figure 1) despite similar surface levels of αv integrin points to particularly important role(s) for Kindlin-2 in αv integrins-VN adhesion and signalling (Liao et al., 2015).

9) Considering the variability in the AFM measurements, it appears that in Figure 2, in contrast to Kind^Ko^ cells, the Tln^Ko^ cells do develop measureable adhesions to FN and that Mn^2+^ increases the strength of the adhesions of nearly control levels. Similarly, Mn^2+^ increased the adhesion of the Kind^Ko^ cells, although less so than the control cells. Since there is evidence that the interaction of integrins with ligands is a multi-step process with adhesion strength increasing as the time of contact between integrin and ligand increases, the interpretation of the these experiments, i.e., the kindlin stabilizes integrin ligand complexes with time by inducing or maintaining integrin clusters is not necessarily the only conclusion.

Indeed, our experiments measure that Tln^Ko^ and Kind^Ko^ cells in the presence of Mn^2+^ restore the time-dependent adhesion strengthening. Whereas the adhesion of Tln^Ko^ cells is fully restored in the presence of Mn^2+^, that of Kind^Ko^ cells is restored by about only 50% (Figure 2). Thus, we conclude that both talin and kindly play distinguishable roles in establishing integrin adhesion and adhesion strengthening. We thank the reviewers for this comment, which encouraged us to discuss the role of kindlin in establishing and strengthening integrin-mediated cell adhesion to FN in more detail. It is possible that kindlin is not only inducing integrin clustering but also modulating the off-rate of integrin-ligand bonds. This role could be mediated through ILK-Pinch-Parvin (IPP) binding, which may link kindlin to the F-actin cytoskeleton. We changed the text accordingly to provide this alternative conclusion.

*10)*
Figure 3: If we have interpreted the distribution of dots correctly, these data suggest that talin and kindlin are not necessarily required to form integrin clusters.

As mentioned in the manuscript, a fraction of kindlin- and almost all talin-deficient cells fail to undergo isotropic spreading. These cells are small, rounded, have small finger-like protrusions and display similar distributions and aggregations of integrin localizations on their surface. It could well be that neither talin nor kindlin is required for clustering. However, these small, non-spread cells may cluster their integrins simply because of a lack of space on the cell surface. We changed the text to make this point clearer.

11) If the control fibroblasts are maintained in suspension, are kindlin and talin constitutively associated with β1 tail? Do the cells constitutively bind 93G7? What is the distribution of integrin in the suspended cells?

a) Since the affinity of talin or kindling to integrin tails is very low, it is difficult or almost impossible to determine the association of talin or kindlin to integrins by regular immunoprecipitation in cell lysates. We therefore performed β1 integrin immunoprecipitation experiments after DSP crosslinking of adherent and suspension cells and used β1 YY/AA expressing cells that carry point mutations in the talin and kindlin binding sites as control. We frequently detect reduced co-IPed kindlin-2 and talin which would argue against a constitutively association with β1 integrin tail. However, the results varied depending on the DSP crosslinking time and would require more time and alternative assays to fully address this issue. As the conclusions in our manuscript do not depend on the question if kindlin-2 and talin are constitutively associated with β1 integrin we did not include this data into the revised manuscript.

Author response image 2.Talin and kindlin binding to the β1 integrin cytoplasmic domain.Kind^Ctr^ or cells expressing a talin and kindlin-binding deficient β1 integrin (β1 YY/AA) were either kept in suspension (susp.) for 3h or seeded on FN-coated dishes (adh.) before incubation for 15 min with 0.5 mM DSP crosslinker at RT. β1 integrin complexes were immunoprecipitated and analyzed by western blotting for presence of talin and kindlin-2.**DOI:**
http://dx.doi.org/10.7554/eLife.10130.030

b) Immunofluorescence staining for β1 integrin in the presence and absences of Mn^2+^, of suspended Tln^Ctr^ and Kind^Ctr^ cells revealed aggregates of β1 integrins in F-actin-positive membrane spikes, which were indistinguishable from Tln^Ko^ or Kind^Ko^ cells. We cannot distinguish between specific vs spatially-driven aggregation although the small cell shapes suggest that they were induced by spatial constraints rather than specific signalling.

Author response image 3.Distribution of β1 integrin in the suspended cells.Immunofluorescence staining for β1 integrin in the presence and absences of Mn^2+^ of suspended Tln^Ctr^ and Kind^Ctr^ cells. Cells were co-stained with phalloidin to visualize F-actin.**DOI:**
http://dx.doi.org/10.7554/eLife.10130.031

[Editors' note: further revisions were requested prior to acceptance, as described below.]

Specifically, the loading controls for Erk, Akt and Cas in kindlin control and KO cells show very variable levels of expression of these proteins in the different cells. Please provide a new gel to address this minor issue.

We have redone the western blots shown in Figure 5—figure supplement 3(b). The variable levels of the control loadings were due to the repeated stripping and re-probing of the membranes. When this is avoided the controls look perfectly fine, as shown in the revised figure. Furthermore, we complemented our western blots showing data from cells that were not treated with Mn^2+^ and/or EGF with a “– – –” above each lane.